

**A Comparison of South Pacific Antarctic Sea Ice and Atmospheric Circulation**
**Reconstructions Since 1900**
Ryan L. Fogt[1], Quentin Dalaiden[2], and Gemma K. O'Connor[3]
[1]Department of Geography and Scalia Laboratory for Atmospheric Analysis, Ohio University,
Athens, OH, USA
[2]Université catholique de Louvain (UCLouvain), Earth and Life Institute (ELI), Louvain-la-
Neuve, Belgium
[3]Department of Earth and Space Sciences, University of Washington, Seattle, WA, USA
**Corresponding Author Address:** Dr. Ryan L. Fogt, Ohio University, 122 Clippinger
Laboratories, Athens OH 45701 USA email:fogtr@ohio.edu



## Abstract

18        The recent changes and record minima in Antarctic sea ice extent implore the need for

longer estimates beyond the short satellite observation period commencing near 1979. However,
Antarctic sea ice extent reconstructions based on paleo records and those generated based on
instrumental observations from the Southern Hemisphere midlatitudes are markedly different,
especially prior to 1979. Here, these reconstructions are examined with the goal of
understanding the relative strengths and limitations of each reconstruction better so that
researchers using the various datasets can interpret them appropriately.

Overall, it is found that the different spatial and temporal resolutions of each dataset play

a secondary role to the inherent connections each reconstruction has with its underlying
atmospheric circulation. Several Southern Hemisphere pressure reconstructions spanning the
$20^{th}$ century are thus examined further. There are different variability and trends poleward of
60°S between paleo-based and station-based $20^{th}$ century pressure reconstructions which are
connected to the disagreement between the Antarctic sea ice extent reconstructions examined
here. Importantly, sensitivity experiments based on only coral paleoclimatological records
provide the best agreement between the early pressure reconstructions, suggesting a contributing
role of tropical variability is present in the station-based pressure (and therefore sea ice)
reconstructions, while high latitude ice core information strongly constrains paleo-based
reconstructions (of both pressure and sea ice) near the Antarctic continent. Our results reveal the
greatest consistencies and inconsistencies in available datasets and highlight the need to better
understand the relative roles of the tropics versus high latitudes in historical sea ice variability
around Antarctica.



## 1. Introduction

The climate of Antarctica is very complex and highly variable in space and time, influenced
by unique processes over the Southern Ocean, the surrounding sea ice, and the continent itself
(Jones et al., 2016; Goyal et al., 2021; Raphael et al., 2016; Holland et al., 2022; Blanchard-
Wrigglesworth et al., 2021).  Although geographically remote from other continents in the
Southern Hemisphere, it is also strongly modulated by large-scale patterns of climate variability,
including teleconnections from the tropical oceans, in particular in West Antarctica (Li et al.,
2021; Lachlan-Cope and Connolley, 2006; Ding et al., 2011).  Some of the more pronounced and
unique changes in Antarctic climate include a rapid acceleration and thinning of outlet glaciers in
the Amundsen Sea embayment (near West Antarctica) (Rignot et al., 2019, 2013; Bamber et al.,
2009; Smith et al., 2020), a strengthening of the atmospheric circulation (in particular in austral
summer) since 1980 linked to stratospheric ozone depletion (Polvani et al., 2011; Banerjee et al.,
2020), and record low total Antarctic sea ice extent set in 2017 (Turner et al., 2017), 2022
(Turner et al., 2022; Wang et al., 2022), and 2023, following multiple decades of a slow
equatorward growth in the sea ice edge from 1979-2016 (Parkinson, 2019; Hobbs et al., 2016).
While the high degree of interannual variability makes it challenging to fully understand
these processes and changes, knowledge of them is also compromised by the comparatively
temporally short length of instrumental observations (Jones et al., 2016).  The majority of
Antarctic meteorological measurements of temperature, pressure, and wind extend back until the
International Geophysical Year (1957-1958), giving roughly only 60 years of continuous
measurements for much of the continent (although most are not located in the interior of the ice
sheet) (Turner et al., 2004, 2020).  The observational record for Antarctic sea ice is even shorter



– beginning near 1978-1979 when modern satellites provided continuous measurements of the
sea ice concentration surrounding the continent (Meier et al., 2021; Parkinson, 2019).

Given the large year-to-year variability and the short observational records, other approaches

must be employed in order to place the shorter records into a longer term context – a necessary
step to better understand the potential uniqueness of ongoing changes across the continent and to
provide more aid in deciphering possible mechanisms for these changes.  One common approach
is to produce reconstructions of past climate prior to direct observational measurements.  For
Antarctica, these reconstructions generally come in two main approaches. The first approach
relies on paleoclimate records such as proxies from ice cores (typically the water isotopic content
or snow accumulation) or ocean sediments to provide estimates of climate back centuries to
millennia (Thomas et al., 2019; Steig et al., 2013; Thomas et al., 2017; Stenni et al., 2017).
These paleoclimate data can be used directly to provide reconstructions of past climate if there is
a strong physical relationship between the paleoclimate data and some aspect of observed
climate; they can also be assimilated with climate model simulations to provide a more spatially
complete reconstruction of climate across Antarctica (Dalaiden et al., 2021a; O'Connor et al.,
2021, 2023).  The second approach is based on instrumental observations in regions away from
Antarctica through statistical models connecting the Antarctic climate with the climate across the
Southern Hemisphere (Fogt et al., 2016a, b, 2019, 2022a; Fogt and Connolly, 2021). Assuming
these relationships are stationary in time (Clark and Fogt, 2019), this approach creates
reconstructions throughout the length of other Southern Hemisphere climate observations based
on the relationship during the period of their overlap with Antarctic climate observations.

The proxy-based and instrumental-based approaches have different strengths and

weaknesses. Paleo-based reconstructions can provide historical changes on longer timescales and



in key regions distant from stations; however, they are often limited to annual resolution and are
associated with uncertainties in the precise climate signals recorded. Station-based
reconstructions can produce sub-annual resolutions with direct observations of climate but are
restricted temporally and spatially by station availability. Additionally, paleo-based
reconstructions have the advantage to rely on measures of Antarctic climate variability that are
located closer to the Southern Ocean, whereas station-based reconstructions largely rely on data
from the Southern Hemisphere midlatitude land masses.  These differences can lead to different
estimates of Antarctic climate in the early 20$^{th}$ century (Fogt et al., 2022b).  In particular, one key
area of differences suggested in earlier work between these various reconstructions was in the
south Pacific and Atlantic Oceans stretching from the Ross Sea east to the Weddell Sea – regions
of strong trends from observations (or reanalyses) in both sea ice concentration / extent and sea
level pressure (Fig. 1).  Recently, Fogt et al. (2022b) discussed this area as a key region where
further analysis is needed on the similarities and differences between the reconstructions to better
understand their utility.  This paper will extend the preliminary analysis of Fogt et al. (2022b) to
provide further comparison between proxy-based and instrumental-based reconstructions of 20$^{th}$
century Southern Hemisphere sea ice extent to better understand the origins of their
discrepancies, with a particular focus on the role of the atmospheric circulation throughout the
20$^{th}$ century as a primary mechanism for these differences.

## Annual Mean Trends 1979-2020

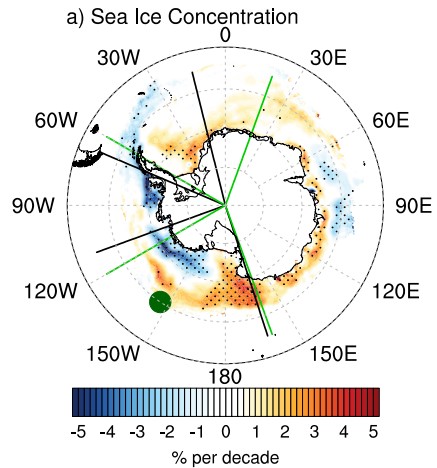

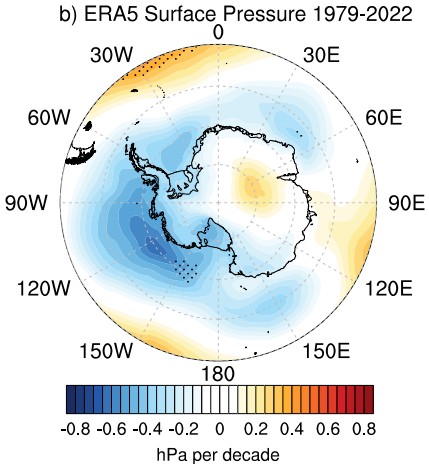

**Figure 1.** a) Annual mean sea ice concentration trends (% per decade) over 1979–2020. The solid black lines denote the sea ice sector boundaries from Parkinson (2019), while the solid green lines represent the boundaries from Raphael and Hobbs (2014). The dark green dot marks the location of the ice edge approximated by the Thomas and Abram (2016) reconstruction. b) Annual mean ERA5 surface pressure trends (hPa per decade). In both plots, trends statistically different from zero at $p<0.05$ are stippled.





## 2. Data and Methods

Table 1 provides further info on the various reconstructions compared in this paper, while Fig. 2 shows the locations of data used in three separate reconstructions. For station-based reconstructions, we use the Fogt et al. (2022a) seasonal sea ice extent reconstructions and the Fogt and Connolly (2021) merged pressure reconstructions. We also investigate three proxy-based reconstructions for sea ice extent (one of which is spatially complete), and two spatially complete, proxy-based reconstructions for atmospheric pressure.

*2.1) Station and sea ice observations and station-based reconstructions*

Monthly mean atmospheric pressure and temperature observations across the Southern Hemisphere used in the Fogt et al. (2019) pressure and Fogt et al. (2022a) sea ice reconstruction (Fig. 2a) are primarily obtained from the University Corporation for Atmospheric Research (UCAR) Research Data Archive dataset ds570.0 (https://rda.ucar.edu/datasets/ds570.0/#!description). A few stations from this dataset have been patched and merged with nearby stations to provide the most complete and continuous observational record, following Fogt et al. (2022a). Some other station data have been corrected independently, as discussed in Fogt et al. (2022a).



**Table 1.** Details on the various datasets used in this study.

| Dataset | Type (station or paleo based) | Variable(s) used in this study | Data Used in Reconstruction | Time Period | Resolution (Temporal / Spatial) |
|---|---|---|---|---|---|
| **Fogt et al. (2022a)** | Station | Sea ice extent | Station pressure and temperature, indices of climate variability (Fig. 2a) | 1905-2020 | Seasonal, 5 sectors (Fig. 2a) + total sea ice extent |
| **Fogt and Connolly (2021)** | Station | Near surface pressure | Fogt et al. (2019) reconstruction merged with NOAA 20CRv3 equatorward of 60°S | 1905-2016 | Seasonal, global, interpolated to 1°x1° |
| **Abram et al. (2010)** | Paleo | Sea ice extent | 3 Antarctic Peninsula ice cores | 1900-2004 | Winter (August-October), 70°W – 100°W (Bellingshausen Sea, Fig. 1a) |
| **Thomas and Abram (2016)** | Paleo | Sea ice extent | Coastal West Antarctica ice core | 1702-2010 | Annual, ice edge at 146°W (Fig. 1a) |
| **Dalaiden et al. (2021)** | Paleo data assimilation with climate model prior | Sea ice extent (derived from sea ice concentration); mean sea level pressure | Ice core $\delta^{18}O$ and surface accumulation; tree ring width data (Fig. 2b); used isotope-enable CESM1 last millennium ensemble as a prior | 1800-2000 | Sea ice extent: Annual, 5 sectors as in Raphael and Hobbs (2014) and Fogt et al. (2022a) Pressure: annual, global, 1°x1° |
| **O'Connor et al. (2021)** | Paleo data assimilation with climate model prior | Mean sea level pressure | Global ice core, tree ring, and coral / sclerosponges (Fig. 2c); used CESM1 Pacific pacemaker ensemble (with external forcings) as a prior | 1900-2005 | Annual, global, 1°x1° |
| **O'Connor et al. (2023)** | Paleo data assimilation with climate model prior | Mean sea level pressure from sensitivity experiments | 1 reconstruction using only ice core data and 1 reconstruction using only coral data | 1900-2005 | Annual, global, interpoated to 1°x1° |



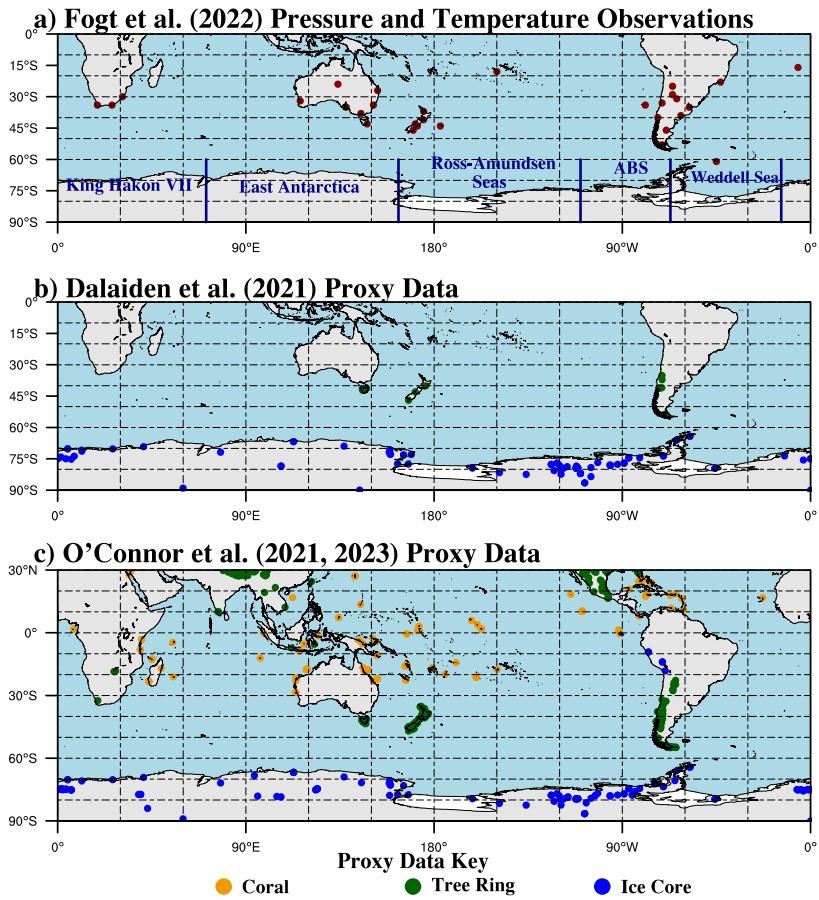

**Figure 2.** Map of a) temperature and pressure observations used in the Fogt et al. (2022a)
seasonal sea ice reconstructions (the same pressure stations were used in the Fogt et al. (2019)
and Fogt and Connolly (2021) pressure reconstruction); b) proxy data locations used in the
Dalaiden et al. (2021) pressure and sea ice reconstructions; c) proxy data locations used in the
O'Connor et al. (2021, 2023) pressure reconstructions. The Fogt et al. (2022a) seasonal sea ice
reconstructions used only a subset of the available observations and also indices of atmospheric
and oceanic variability in their reconstructions, depending on the sea ice sector being
reconstructed (Table 1), depicted in a) (ABS= Amundsen-Bellingshausen Seas). For b), the ice
core locations are a combination of the $\delta^{18}$O and the surface accumulation measurements. For
c), the coral proxies also include sclerosponges. Although the O'Connor et al. (2021, 2023)
assimilated proxy data span the entire globe, only the data south of 30°N are shown as these have
the strongest influence on the reconstruction near Antarctica.



Observed sea ice concentration, from which sea ice extent is calculated, is obtained from the
Nimbus-7 Scanning Multichannel Microwave Radiometer (SMMR) and Defense Meteorological
Satellite Program (DMSP) Special Sensor Microwave Imager - Special Sensor Microwave
Imager/Sounder (SSM/I-SSMIS).  The Fogt et al. (2022a) sea ice extent reconstructions use the
Climate Data Record (CDR) daily concentration fields from the National Oceanic and
Atmospheric Administration / National Snow and Ice Data Center (NOAA/NSIDC) Climate
Data Record of Passive Microwave Sea Ice Concentration, Version 4
(https://nsidc.org/data/g02202) (Meier et al., 2021).  The CDR algorithm output combines of ice
concentration estimates from the National Aeronautics and Space Administration (NASA) Team
algorithm and the NASA Bootstrap algorithm, and are available at a 25 km x 25 km polar
stereographic grid.  Sea ice extent is calculated as the equatorward limit of the area bounded by
15% sea ice concentration isoline; monthly, seasonal, and annual means are calculated from the
daily sea ice data. Patching of a short temporal discontinuity of the sea ice observations between
December 1987 and January 1988 was done as in Fogt et al. (2022a).  Longitude bounds for the
sea ice sectors used in this study follow Raphael and Hobbs (2014), specifically defined as:
Amundsen-Bellingshausen Seas (250°-290°E), Weddell Sea (290°-346°E), and the Ross-
Amundsen Sea (162°-250°E); see Figs. 1 and 2a for boundaries.
The seasonal spatially complete pressure reconstructions of Fogt et al. (2019) span the region
poleward of 60°S at 100km resolution, and are a kriging interpolation of individual Antarctic
station reconstructions from Fogt et al. (2016a,b).  The Fogt et al. (2019) seasonal Antarctic
pressure reconstruction has been merged with the National Oceanic and Atmospheric
Administration (NOAA) 20[th] century reanalysis version 3 data (Slivinski et al., 2019)



equatorward of 60°S, as discussed in Fogt and Connolly (2021), which we use for comparison in
this study.
*2.2) Paleo-based reconstructions*
While there are a few ocean sediment derived sea ice extent reconstructions near the
Antarctic Peninsula relevant to this study, Thomas et al. (2019) note that the poorer temporal
resolution and lack of calibration of most marine sediments create challenges when combining
and comparing them to ice core derived sea ice reconstructions.  We therefore employ two main
ice core based reconstructions of sea ice extent, as in Fogt et al. (2022b): Abram et al. (2010)
used a three ice cores from the Antarctic Peninsula to reconstruct the winter sea ice extent from
70°W – 100°W, in the Bellingshausen Sea (outlined in Fig. 1a).  The Abram et al. (2010)
reconstruction ends in 2004.  Further west, Thomas and Abram (2016) provide a reconstruction
with annual resolution of the sea ice extent in the Ross Sea (marked in Fig. 1a). Both Abram et
al. (2010) and Thomas and Abram (2016) reconstructed the sea ice extent based on the
methanesulphonic acid (MSA) content – an indicator related to the algal blooms occurring
during the ice break-up periods -  from different Antarctic ice cores by calibrating them against
sea ice extent from satellite observations.
As mentioned earlier, multiple ice core, tree-ring, and coral proxy records can be assimilated
with Earth System Model simulations to provide annual mean gridded estimates of not only sea
ice extent and concentration, but also atmospheric pressure.  We employ two such estimates of
previous Antarctic climate in this study- those of Dalaiden et al. (2021) and O'Connor et al.
(2021, 2023; no sea ice reconstruction), which both use proxy measurements but different data
assimilation filters and different Earth System Model simulations as the data assimilation "prior"
(the initial guess that is updated with proxy data) (Table 1). Although no sea ice nor atmospheric



pressure observations are directly assimilated, the data assimilation relies on the covariance
between assimilated observations and those variables given by the Earth System Model to
reconstruct them (Widmann et al., 2010; Goosse et al., 2010; Hakim et al., 2016). The temporal
variability thus only comes from proxy records, which are spatially interpolated during data
assimilation. Therefore, the final reconstruction is dynamically consistent through all the
reconstructed climate variables.  For Dalaiden et al. (2021) we use the sea ice concentration,
extent, and sea level pressure datasets generated using the isotope-enabled Community Earth
System Model version 1 (CESM1) last millennium ensemble as the prior (Brady et al., 2019),
and based on Southern Hemisphere ice cores and tree rings (Fig. 2b). We further use the
O'Connor et al. (2021) pressure dataset generated using the CESM1 tropical Pacific pacemaker
ensemble of simulations (Schneider and Deser, 2018) as the data assimilation prior, as this
ensemble best represents historical external forcing and Pacific variability relative to other
simulations (for more details see O'Connor et al., 2021). The O'Connor et al. (2021) dataset
includes a global proxy database of ice cores, corals, and tree rings, synthesized by the
PAGES2k working group (Fig. 2c) (PAGES2k Consortium et al., 2017), with additional snow
accumulation records (Thomas et al., 2017).  The comparison between the Dalaiden et al. (2021)
reconstruction and the O'Connor et al. (2021) reconstruction reveals the roles of different
filtering methods, proxy databases, and climate model priors.  O'Connor et al. (2023) provide
additional single-proxy reconstructions, using the same configuration as O'Connor et al. (2021),
but based only on the assimilation of ice core or coral records.  The comparison between these
sensitivity experiments with O'Connor et al. (2021) allows a better understanding of the possible
role of certain proxy data on the resulting reconstruction.





**3. Results**
*3.1) Comparison of Antarctic sea ice extent reconstructions*
As alluded to in the introduction and Fogt et al. (2022a,b), there are substantial differences
between the station-based sea ice extent reconstructions of Fogt et al. (2022a) and those based on
paleo data, including from data assimilation-based reconstructions. To investigate these
differences further, the various time series of standardized (to place on same scale) annual-mean
sea ice extent from the Ross-Amundsen Sea sector east to the Weddell Sea (see Fig. 1 for sector
boundaries) are plotted in Fig. 3. Not surprisingly, the correlations with the Fogt et al. (2022a)
reconstructions with the observed data (green number in upper right of each panel) are the
highest ($p<0.01$), as these reconstructions were specifically statistically calibrated to provide the
best match to the observations. However, the correlations of the paleo-based (including the
Dalaiden et al. (2021) reconstruction which is not directly calibrated to sea ice observations)
reconstructions are also uniformly positive, and the Thomas and Abram (2016) provides the
highest correlations of the paleo-based reconstructions in the Ross-Amundsen Seas (Fig. 3a,
r=0.525, $p<0.01$), while the Dalaiden et al. (2021) estimates from their data assimilation-based
reconstruction are the highest correlations nearer to the Antarctic Peninsula in the
Bellingshausen-Amundsen (Fig. 3b, r=0.620, $p<0.01$) and Weddell Seas (Fig. 3c, r=0.445,
$p<0.01$).



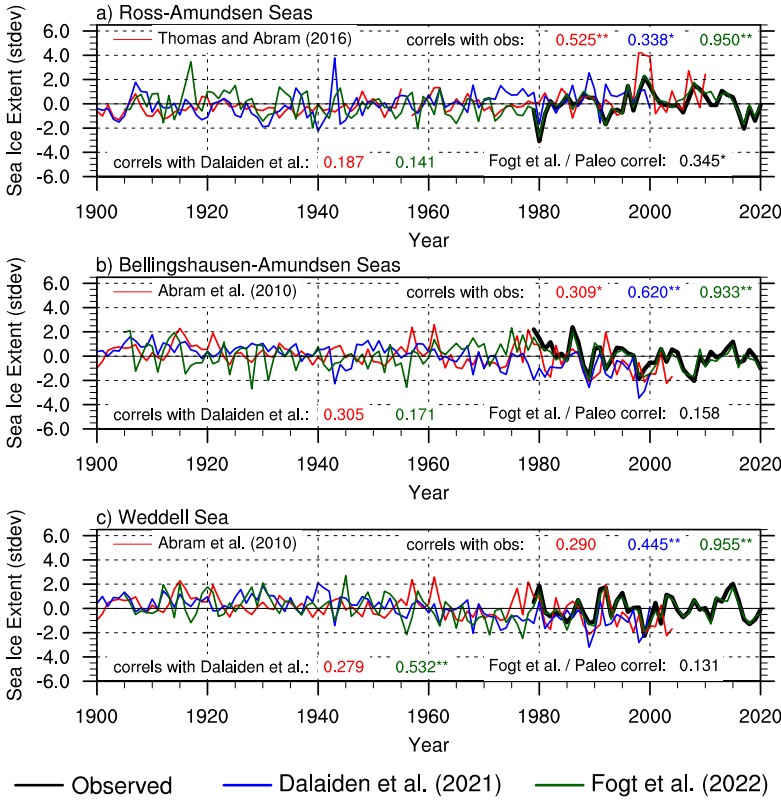

**Figure 3.** Annual mean sea ice extent timeseries from various reconstructions approximately
related to the Raphael and Hobbs (2014) sea ice boundaries in Fig. 1 for the a) Ross-Amundsen
Seas; b) Bellingshausen-Amundsen Seas; c) Weddell Sea. The Dalaiden et al. (2021) sectors
have been adjusted to the Raphael and Hobbs (2014) boundaries. The correlations with satellite
observations (in black) during the period of overlap are given in the top of each panel, with the
colors denoting the individual timeseries. At the bottom, correlations with the Dalaiden et al.
and Fogt et al. reconstructions are similarly provided. In each case, after adjusting the sample
size by the lag1 correlation as needed, correlations significantly different from zero at $p<0.05$
and $p<0.01$ are marked with * and **, respectively.

In contrast, however, the cross-correlations between the various sea ice extent

reconstructions are much weaker overall (numbers at bottom of each plot), often falling below

0.30 (from 1905-2020, $p<0.05$), suggesting that the interannual variability in the Dalaiden et al.

(2021) estimates from a climate model that assimilates paleo data and estimates from ice cores

directly (both Thomas and Abram (2016) and Abram et al. (2010)) differ substantially; part of



this difference could be from a larger number of ice cores included in the Dalaiden et al. (2021)
reconstruction (Fig. 2b) than those of Abram et al. (2010) and Thomas and Abram (2016).  In the
Bellingshausen-Amundsen sector (Fig. 3b), even though the Dalaiden et al. (2021) estimate
shows a correlation of r=0.620 with observations (blue number, top of plot), the correlation of
the Fogt et al. (2022a) data with the Dalaiden et al. (2021) estimate is only r=0.171 ($p$>0.05)
during 1905-2000.  Since the Fogt et al. (2022a) data are strongly correlated r=0.933 ($p$<0.01)
with observations after 1979, the low correlation between these two estimates during 1905-2020
suggests they do not have similar interannual variability prior to 1979, as demonstrated in Fogt et
al. (2022a,b).  Of note, though, are a few cross-correlations that are near the same strength of the
paleo-based estimates with observations, namely the Abram et al. (2010) correlation with the
Dalaiden et al. (2021) estimates in the Bellingshausen-Amundsen Sea (Fig. 3b, r=0.305, $p$>0.05)
and the Fogt et al. (2022) and Dalaiden et al. (2021) estimates in the Weddell Sea (Fig. 3c,
r=0.532, $p$<0.01).  Even though there are some agreements, the overall linear trends in the data
look notably different (to be addressed later), and there are sudden anomalies in each dataset that
are rarely replicated in others, including for the paleo-based reconstructions in the period of
satellite observations.

While it is straightforward to plot the timeseries together as in Fig. 3, understanding the

differences between them is much more complex, as the various reconstructions were created in
markedly distinct and incongruent methods (Table 1) and data (Fig. 2).  Further, while they are
all representing sea ice extent in some fashion, the reconstruction resolutions (Table 1) also
indicates they are representing it in many different ways as well (some as the ice edge latitude at
a specific point, some as an area, and all with different temporal resolutions).  One important
challenge is that the Fogt et al. (2022a) reconstructions were generated based on seasonal



statistical relationships, producing reconstructions separately for the four meteorological seasons.
These seasonal reconstructions were averaged together to give the annual mean values in Fig. 3
to compare with all the other paleo-based reconstructions that only have annual mean resolution
(the exception being the MSA-based reconstruction of Abram et al. (2010) is for August-
October, Table 1).  Nonetheless, even though the paleo-based reconstructions often represent the
annual mean, they may be biased slightly to a particular time in the year affected primarily by the
accumulation at the ice core site(s) from which the paleo-based reconstructions were generated.
To understand if the relationships improve seasonally, Fig. 4 investigates the reconstructions
agreement for each sector (by column) and each season (rows).
Compared to the annual mean data in Fig. 3, there are frequent negative correlations
between the various datasets and the observations (again except for the Fogt et al. (2022a)
reconstructions that were explicitly calibrated to the seasonal observations), as well as between
the paleo-based reconstructions and those from Fogt et al. (2022a) given by the numbers at the
bottom of each panel in Fig. 4.  For the Ross-Amundsen sector, the Thomas and Abram (2016)
most closely aligns with the austral winter (JJA) and austral spring (SON) seasons, near the
seasonal maximum sea ice extent.  The correlations with observations of the Thomas and Abram
(2016) data exceed 0.56 ($p<0.01$) in these seasons, higher than the correlation of the annual mean
in Fig. 3 (r=0.525, $p<0.02$); despite the higher correlations with the observations, the correlations
of Thomas and Abram (2016) and the seasonal Fogt et al. (2022a) reconstructions however are
slightly lower than for the annual (r=0.345 for annual, max for SON is r=0.320, both $p<0.05$).
As before, this suggests a reduction in the reconstruction cross correlations prior to satellite
observations (i.e., before 1979).  The seasonal relationships of the Abram et al. (2010)
reconstruction are a bit more nuanced: it has considerably higher correlations in MAM with


observations in the Bellingshausen-Amundsen sector (r=0.526, *p*<0.01), but the weakest
correlations in this season in the Weddell sector (r=0.10, *p*>0.05).

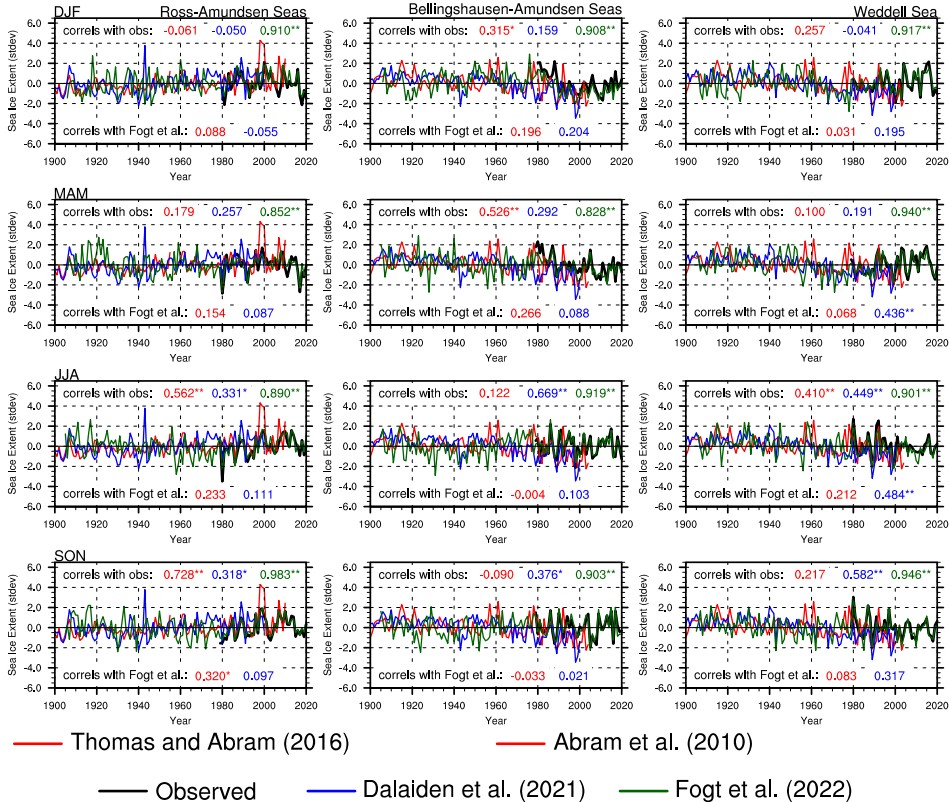

**Figure 4.** As in Fig. 3, but with the seasonal mean observations and reconstructions of Fogt et
al. (2022) with the annual mean paleo-based reconstructions. The Thomas and Abram (2016)
paleo-based reconstruction is used for the Ross-Amundsen sector (left column), while the Abram
et al. (2010) reconstruction is used for the other two sectors.

For the Dalaiden et al. (2021) estimates of sea ice from the data assimilation-based
reconstruction, the relationships with observations are weakest in austral summer (top row) for
all sectors, and generally the highest in austral winter and spring (bottom two rows).
Interestingly, the correlations between the Fogt et al. (2022a) reconstruction and the estimates
from Dalaiden et al. (2021) are in typically highest in JJA, and even exceed the correlations of



the paleo-based reconstructions with observations in the Weddel sector in this season (r=0.484,
$p<0.01$), suggesting that there is some shared interannual variability in these datasets prior to
1979 in the winter season.

While the seasonal comparisons reveal that better agreement between the various

reconstructions can be achieved apart from the annual mean representation, there are other
factors that could lead to differences in the reconstruction that would not be captured by
individual portions of the seasonal cycle.  In particular, while ice core-based reconstructions can
provide information directly over the continent on longer timescales, their connections to
Antarctic climate are geographically limited (Table 1), restricted to the prominent pathway of
tracers from the ocean/ice boundary near the ice edge to their deposition at the ice core site
(Thomas et al. 2019).  In contrast, both the Fogt et al. (2022a) and Dalaiden et al. (2021)
reconstructions represent the cumulative sea ice area >15% in specific geographic boundaries
(Figs. 1a and 2a).  To investigate the role the various spatial configurations may play in the
differences between the reconstructions, each annual mean reconstruction was correlated through
time with the full spatial field of annual mean sea ice concentration satellite observations (at each
grid point separately); these correlations are plotted in Fig. 5, with correlations statistically
different from zero at $p<0.05$ stippled.  In Fig. 5, the correlation of the observed sea ice extent
series for each sector with the satellite sea ice concentration field is provided in the far left
column for comparison of the maximum expected correlation pattern and magnitude.



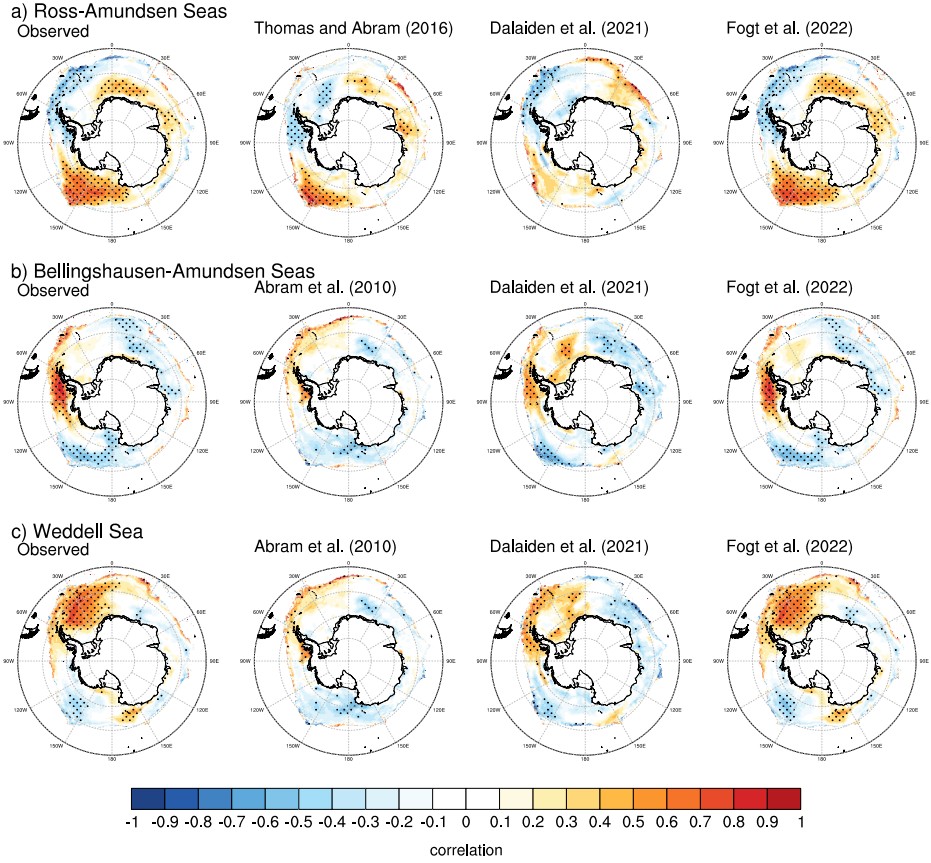

**Figure 5.** Correlations (1979-end of series) of annual mean sea ice extent timeseries with the
observed annual mean sea ice concentration field, separated by the Raphael and Hobbs (2014)
sectors, as in Fig. 3. In each row, correlations of the observed sea ice extent time series with the
sea ice concentration are given for reference. a) Ross-Amundsen Seas; b) Bellingshausen-
Amundsen Seas; c) Weddell Sea.


In the Ross-Amundsen sector (Fig. 5, top row), the Thomas and Abram (2016) and Fogt et
al. (2022a) reconstructions represent the observed pattern well, although as expected the region
of positive correlations with the Thomas and Abram (2016) is smaller and confined closer to the
ice edge – their reconstruction is an estimate of the ice edge at a point in the Ross Sea (Fig. 1,
Table 1). While the Dalaiden et al. (2021) spatial correlation field is notably weaker, it is
important to remember that unlike the other sea ice extent estimates examined here, this





reconstruction was not calibrated in any way to sea ice, and was only extracted from the climate
model simulations constrained by paleo data. Nonetheless, it still maintains a similar spatial
pattern of positive and negative correlations to the other datasets, suggesting that its geographic
representation is broadly similar to other datasets.
The Abram et al. (2010) reconstruction shows a much weaker pattern of correlations than
the other estimates in the Bellingshausen-Amundsen and Weddell sectors (middle and bottom
rows of Fig. 5), while the Dalaiden et al. (2021) estimates align much better with observations in
these sectors. The Abram et al. (2010) relationships with the sea ice concentrations only
modestly improves seasonally (not shown). Together, this suggests that the spatial (and to a
lesser extent, seasonal) limitation of the ice conditions represented by the Abram et al. (2010)
reconstruction is an important contributing factor of its differences with the other sea ice extent
estimates examined here. From Fig. 5, the Abram et al. (2010) reconstruction has significant
correlations at the far equatorward sea ice edge, stretching from the Bellingshausen Sea eastward
across the Antarctic Peninsula to the Weddell Sea, different than the 70°W-100°W region
originally suggested. This different spatial relationship in the annual mean (compared to the
originally published August – October) also explains why it has correlations with sea ice
observations from both the Bellingshausen-Amundsen and Weddell sectors (Figs. 3-4), albeit
much weaker than other reconstructions.
The timeseries in Figs. 3-4 suggest that while there are notable differences in the interannual
variability in the various reconstructions, there are also differences in longer-term changes and
especially the trends through time. Figure 6 investigates these discrepancies through the use of
30-year running trends, similar to that done in Fogt et al. (2022b), but including the Weddell
sector and other data sources.



**Figure 6.** 30-year running trends (standard deviations per decade) of the annual mean sea ice extent timeseries in Fig. 3. Individual 30-year trends that are statistically different from zero at $p<0.05$ are stippled. For the Weddell Sea, the fast ice duration at the Orkney islands from Murphy et al. (1995) is also provided. For the Bellingshausen-Amundsen Sea region, the ASL index from Dalaiden et al. (2021), defined as the annual mean 500 hPa geopotential heiht from 60°-75°S, 170°E – 70°W, is also shown.

One clear pattern in the satellite-observed trends discussed in the introduction and shown in Fig. 1a is the opposing nature of positive sea ice extent trends (equatorward movement of the ice edge) in the Ross Sea and negative sea ice extent trends in the Bellingshausen-Amundsen sector (Fig. 6). As suggested in Fogt et al. (2022a) in their investigation of the total Antarctic sea ice extent trends and evident in Fig. 6, these trends reverse and weaken after the 2016 sudden decline, but still demonstrate opposite behavior, strongly tied to the atmospheric circulation
around the Amundsen Sea low (Hosking et al., 2013; Raphael et al., 2016; Holland and Kwok,
2012).  While the various reconstructions capture the observed trends (including their statistical
significance), there are marked differences in the signs of the trends prior to the satellite
observations around 1979.  In particular, paleo-based sea ice extent estimates generally display
that the observed trends are part of a long-term continuous trend of the same sign throughout
much of the 20th century (i.e., that there are increases in the Ross Sea sector and decreases in the
Bellingshausen Sea region throughout the 20th century); this story is also consistent in the
Weddell sector, as the paleo-based reconstructions only show statistically significant trends prior
to 1979 that are the same sign as the observed trends after 1979.  In contrast, the Fogt et al.
(2022a) reconstructions show a pronounced shift in the trend sign and magnitude through time.
The Fogt et al. (2022a) sea ice reconstruction trends are characterized with statistically
significant positive trends in the middle 20th century in the Bellingshausen-Amundsen sector
(opposite the decreases after 1979 there), and statistically significant negative trends in the Ross
Sea in the early 20th century, with a prolonged period of weaker negative trends throughout the
20th century, opposite the strong positive trends in the satellite observations starting in 1979 (Fig.
6).  In the Weddell sector, where there is a better agreement between the Fogt et al. (2022a) and
Dalaiden et al. (2021) estimates (Figs. 3-5), the trends align better through time.  Moreover, these
trends and their temporal changes in both significance and sign are broadly aligned with the
South Orkney fast ice duration dataset of Murphy et al. (1995), with positive trends in the first
portion of the 20th century that change to negative trends through much of the middle 20th
century, to weak trends during the satellite observation period.

While the seasonal and spatial differences between the various reconstructions evaluated in

Figs. 2-3 undoubtedly play a role (especially for the Abram et al. (2010) reconstruction), Fig. 5





suggests that changes in the underlying implied atmospheric circulation are also playing a role.
In observations (Fig. 1a), the differing sea ice extent trends in the Bellingshausen and Ross Seas
are largely tied to the atmospheric circulation around the Amundsen Sea low and implied sea ice
drift changes (Holland and Kwok, 2012; Holland, 2014).  Although there are no long-term direct
observations, the ASL index extracted from the pressure field in the proxy-based reconstructions
of Dalaiden et al. (2021) show changes through time that are nearly opposite the Fogt et al.
(2022a) reconstructions for the Bellingshausen-Amundsen sector in the early and middle 20th
centuries, potentially confirming that underlying atmospheric circulation changes are a dominant
contribution to the different sea ice changes.
While reconstruction uncertainty will always play a role in differences between various
historical estimates, the comparisons of the various sea ice reconstructions thus far suggest that:
a) the Abram et al. (2010) is primarily different from other reconstructions because of its
different spatial footprint; b) the methodology used to create the reconstructions or their temporal
resolution does not play a consistent role, as correlations between the proxy-based
reconstructions as well as with the station-based reconstructions vary considerably; c) the
atmospheric circulation associated with the sea ice reconstructions appears to be a dominant
mechanism for differences between them. To investigate this further, the role of implied changes
in the atmospheric circulation underlying the proxy-based and station-based reconstructions is
therefore the focus for the remainder of this paper.
*3.2)    Connection to the atmospheric circulation changes and comparison of Antarctic sea level*

*pressure reconstructions*

Since 20th century atmospheric reanalyses have been shown to have long-term artificial
pressure trends throughout the early and middle 20th century (Schneider and Fogt, 2018; Fogt et



al., 2020), our analysis of the relationship between sea ice extent and the atmospheric circulation
is focused on other estimates of pressure. In particular, for consistency we employ estimates of
pressure from reconstructions generated using proxy data assimilation with various climate
model priors and proxy datasets, including the Dalaiden et al. (2021) and O'Connor et al.
(2021,2023) datasets (Table 1). For a station-based estimate, we also investigate the Fogt and
Connolly (2021) dataset, which is a blend interpolated seasonal Antarctic station pressure
reconstructions south of 60°S (Fogt et al., 2019, 2017), and the NOAA 20th century reanalysis
version 3 equatorward of 60°S (Table 1). Since the Antarctic station pressure reconstructions
were generated using a similar statistical technique as the Fogt et al. (2022a) sea ice extent
reconstructions, this allows for an evaluation of other estimates that are expected to provide
similar temporal variability as the Fogt et al. (2022a) sea ice extent reconstructions.

Figure 7 displays the correlations for the Weddell sector sea ice extent from Dalaiden et al.

(2021) with the gridded pressure datasets in the top rows, and the Weddell sector sea ice extent
reconstructions from Fogt et al. (2022a) with the same gridded datasets in the bottom rows. The
gridded pressure datasets are grouped by columns, and further broken into the pre-satellite sea
ice observation period of 1905-1978 (top rows of each section) and satellite-era sea ice
observation period (1979-2000). Overall the patterns are quite similar in each of the panels,
reflecting the broad similarities of the sea ice reconstructions in the Weddell sector (Figs. 3-5).
Here, the sea ice extent reconstructions show a strong positive correlation with pressure over the
Antarctic continent that changes to negative correlations with pressure in the Pacific Ocean from
the midlatitudes and equatorward, including South America. For both the Dalaiden et al. (2021)
and O'Connor et al. (2021) pressure datasets, regardless of the sea ice extent estimate (top or
bottom half of Fig. 7), these relationships exist in a similar fashion throughout time, suggesting

## Weddell Sea SIE Extent and Pressure Correlations

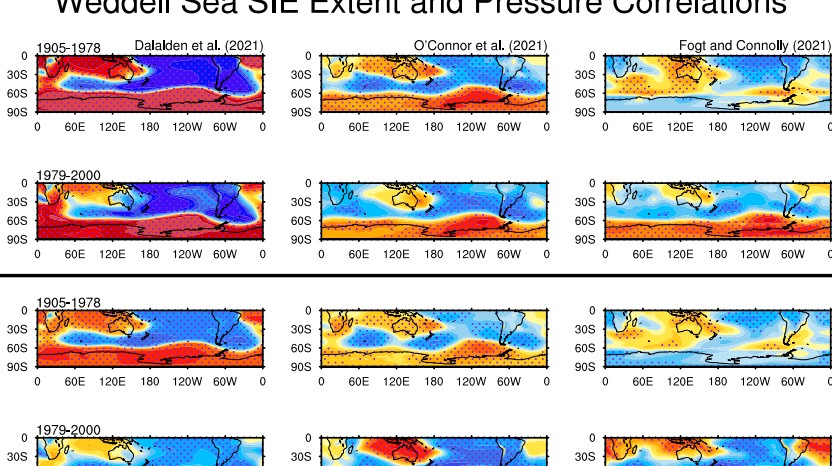

**Figure 7.** Annual mean correlations of (top half) Dalaiden et al. (2021) and (bottom half) Fogt et al. (2022a) sea ice extent correlations for the Weddell Sea with three 20[th] century spatially complete pressure reconstructions (columns). In each half, the top row are for correlations in the pre-satellite period, 1905-1978, and the bottom are for 1979-2000. The pressure reconstructions are (left column) Dalaiden et al. (2021), (middle column) O'Connor et al. (2021), and (right column) Fogt and Connolly (2021). Correlations statistically different from zero at $p<0.05$ are stippled.

the Weddell sea ice maintains a similar relationship with pressure across the entire Southern

Hemisphere from data assimilation-based products with a fixed prior. However, when

comparing with the Fogt and Connolly (2021) pressure dataset, the relationship between Weddell

sea ice extent, regardless of the sea ice extent estimate (Dalaiden et al. (2021) or Fogt et al.

(2022a)) shows a change in the relationship over Antarctica, flipping to a weakly negative

(generally not statistically significant) correlation poleward of 60°S in the 1905-1978 (right

column in Fig. 7). While the trends in the Weddell sea ice extent change less dramatically

through time than in the Ross, Amundsen, and Bellingshausen sectors (Fig. 6), there are still



changes in the sea ice extent trends in the Weddell sector in the Fogt et al. (2022a)
reconstruction, with perhaps too strongly positive sea ice extent trends at the end of the 20[th]
century (compared to observations) juxtaposed by negative sea ice extent trends in the middle
twentieth century (Fig. 6).  If the statistical relationship between the atmospheric circulation and
sea ice extent changed in time (Fig. 7, right column), this could be linked (at least statistically,
and perhaps incorrectly) to the changes in the sea ice extent trends displayed in the Fogt et al.
(2022a) sea ice extent reconstructions, but not in the paleo-reconstructions (Fig. 6).

Interestingly, when examining the correlation of sea ice extent reconstructions in the Ross-

Amundsen sector with the various gridded historical pressure datasets (Fig. 8), this pattern is not
fully maintained, and there are more differences in the spatial correlation patterns not only
between the two sea ice extent estimates, but also dependent on the various pressure datasets
(columns in Fig. 8).  For the Dalaiden et al. (2021) sea ice extent estimate (Fig. 8, left column),
the pressure relationships are maintained throughout time, and are even stronger in the 1905-
1978 period for the Dalaiden et al. (2021) pressure estimate.  As in the Weddell sector, the
correlation pattern of the Dalaiden et al. (2021) sea ice extent estimate with the the Fogt and
Connolly (2021) pressure dataset in the again changes through time poleward of 60°S over the
Antarctic continent, from overall weakly positive and insignificant correlations from 1905-1978
to statistically significant ($p<0.05$) negative correlations after 1979 (Fig. 8, top two rows of
rightmost column).  A notable exception however is off the coast of West Antarctica, in the
vicinity of the Amundsen Sea low, where correlations remain significantly ($p<0.05$) negative
throughout time, highlighting the key role of this feature for regional sea ice variability (Hosking
et al., 2013; Raphael et al., 2016).

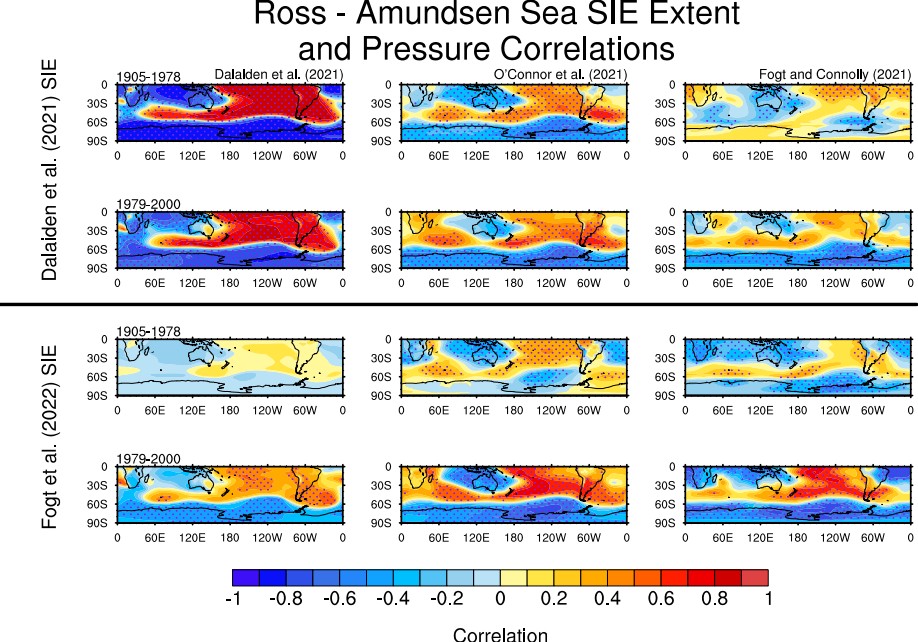

**Figure 8.** As in Fig. 7, but for the sea ice extent reconstructions for the Ross-Amundsen Seas sector.

For the Fogt et al. (2022a) sea ice extent reconstructions (bottom two rows in Fig. 8), a different story emerges. The correlations, while of a generally similar sign, are weaker for all pressure datasets (columns in Fig. 8) in the 1905-1978 period compared to the 1979-2000 period. In particular, the statistically significant ($p<0.05$) negative correlations within the region of the Amundsen Sea low are maintained throughout time, especially in the Fogt and Connolly (2021) pressure dataset (Fig. 8, right column, bottom two rows); these are reduced however when using the Dalaiden et al. (2021) pressure dataset with the Fogt et al. (2022a) sea ice extent reconstruction (Fig. 8, left column, bottom two rows).

Are changes in time of the sea ice – pressure relationship in time in the Weddell sector from Fig. 7 a real feature, or is this an artifact of the reconstructions (sea ice extent, pressure, or both) generated by their limitations? While there are no direct spatially complete observations of sea





ice or pressure throughout the full 20[th] century, the Weddell sector is one of the few places with
longer historical observations of both pressure and sea ice conditions, afforded by the
observations collected at Orcadas station (60.7°S, 44.7°W) since 1903 (Zazulie et al., 2010), and
the fast-ice duration series around this island from Murphy et al. (1995).  The running
correlations of these key observations with sea ice estimates are provided in Fig. 9 as the only
observation-based investigation into the possible reality of a time-varying sea ice / pressure
relationship within the Weddell sea.  In Fig. 9a, the annual mean Fogt et al. (2022a) Weddell sea
ice extent reconstruction maintains a statistically significant ($p<0.05$) positive relationship
throughout time with the South Orkney fast ice (SOFI) record (solid lines, Fig. 9a).  Given this, it
is not surprising then that both the relationship between the Fogt et al. (2022a) Weddell sea ice
extent reconstruction and the SOFI duration similarly change sign through time with the pressure
at Orcadas station (dashed lines in Fig. 9a), potentially confirming the spatial pressure change
through time presented in the right column of Fig. 7.  While the Weddell sea ice extent estimate
from Dalaiden et al. (2021) also is positively correlated through time with the SOFI duration
(solid blue line, Fig. 9b, albeit only significant [$p<0.05$] for 30-year windows starting prior to
1945), the Orcadas observed pressure and Weddell sea ice extent estimate from Dalaiden et al.
(2021) are weakly correlated ($p>0.05$), near zero through time (although still with some
deviation in sign, but not statistically meaningful).  There is therefore much stronger support of
changes in the relationship of high latitude pressure and Weddell sea ice from the statistically-
generated sea ice and pressure reconstructions than there is from the paleo-based reconstructions,
consistent with Fig. 7.



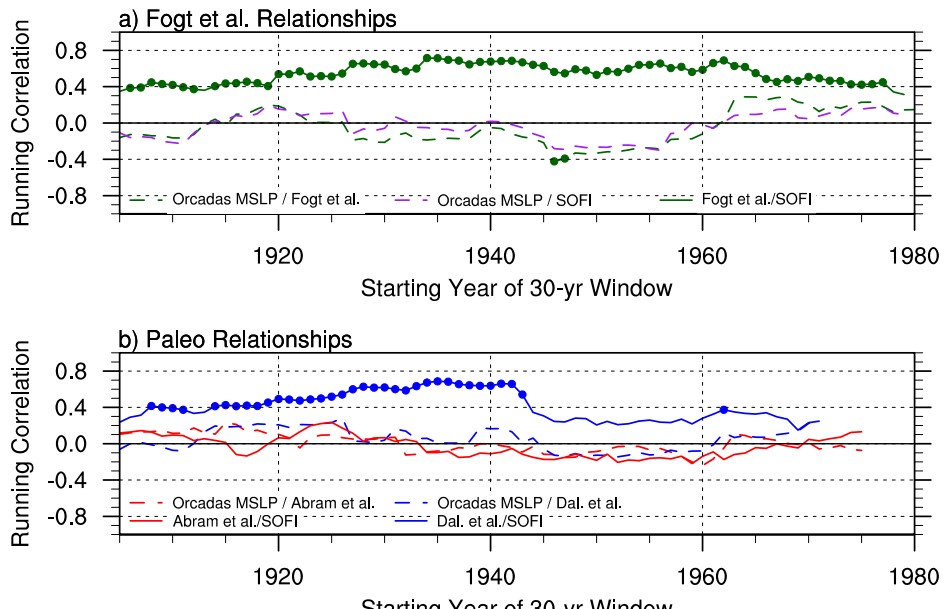

**Figure 9**. 30-year running correlations between pairs of the South Orkney Fast Ice (SOFI) duration from Murphy et al. (1995), various sea ice extent estimates, or Orcadas station mean sea level pressure. Correlations significantly different from zero at $p<0.05$ are marked with a circle.

The connections of the various sea ice reconstructions with the atmospheric circulation in Figs. 7-9 suggest, just like with the sea ice, that there are also differences in the underlying atmospheric circulation from the Fogt and Connolly (2021) dataset and the Dalaiden et al. (2021) and O'Connor et al. (2021) proxy-data assimilation datasets. While some of these differences were discussed in O'Connor et al. (2021), it is not clear how they may impact sea ice changes through time. To understand these differences better and the role they may play in the differences in the sea ice reconstructions, correlations between the various annual mean gridded pressure datasets are provided in Fig. 10. To help pinpoint the sensitivity of the correlations to the paleo data assimilated into the climate model, the O'Connor et al. (2021) pressure reconstruction was expanded by separately examining reconstructions assimilating ice core and coral data only from O'Connor et al. (2023). We examine these separately to better isolate the



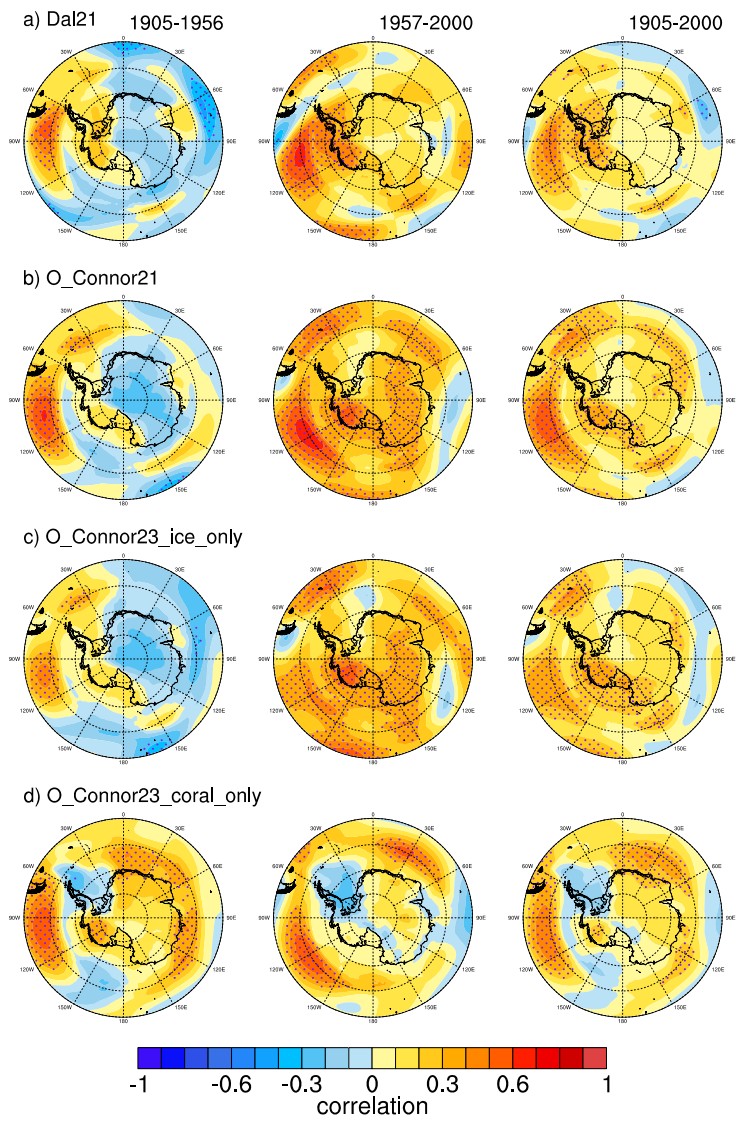

**Figure 10.** Correlations of annual mean pressure at each grid point between the various paleo-based reconstructions with the Fogt and Connolly (2021) reconstruction. a) Dalaiden et al. (2021); b) O'Connor et al. (2021); c) O'Connor et al. (2023) ice core-based proxies only; d) O'Connor et al. (2023) coral-based proxies only. The columns denote three time periods: 1905-1956 (left column, prior to Antarctic data), 1957-2000 (middle column), and the full period of overlap 1905-2000 (right column). Correlations statistically different from zero at $p<0.05$ are stippled. All datasets were linearly detrended over the specific time period prior to calculating the correlations; further details on each dataset is provided in Table 1 and the proxy locations are given in Fig. 2.



role of tropical paleo data constraints (from corals) and those primarily from Antarctica (from ice
cores, Fig. 2c) on the comparison of these pressure reconstructions with the Fogt and Connolly
(2021) data.  In the 1905-1956 time period (left column, Fig. 10), there are similar correlation
patterns between the Dalaiden et al. (2021), O'Connor et al. (2021), and the O'Connor et al.
(2023) ice core only datasets, suggesting that the ice core data provide a strong constraint to the
pressure variability in the paleo-based reconstructions near Antarctica, and that the inclusion of
external forcing in the data assimilation prior (for the O'Connor et al. (2021, 2023)
reconstructions) has minimal influence.  Meanwhile, there are positive and even statistically
significant ($p<0.05$) correlations between the O'Connor et al. (2023) and Fogt and Connolly
(2021) datasets (Fig. 10d, left column) in the early period.  This suggests that tropical
teleconnections, captured by the coral paleo data, play an important role in the interannual
variability in the Fogt and Connolly (2021) dataset.  Given that the predictor data for the Fogt
and Connolly (2021) dataset is confined to the Southern Hemisphere landmasses large distances
away from Antarctica, it is not surprising that there is better agreement with the coral-only
dataset of O'Connor et al. (2023) and Fogt and Connolly (2021) pressure dataset over Antarctica.
However, given the different pattern with the paleo-based datasets (left column, Fig. 10a-c), it is
also clear that the local ice core variability over Antarctica is opposite that of the coral-based
tropical variability (especially over East Antarctica, where tropical teleconnections to Antarctica
are generally weaker; Li et al., 2021), with the former dominating the overall reconstruction for
the paleo-based datasets.  Notably, during 1957-2000 the agreement improves from 1905-1956
for the full reconstructions (Fig. 10a-b, middle column, especially for the O'Connor et al.
reconstructions) as well as the ice core-only reconstruction from O'Connor et al. (2023, Fig. 10c,
middle column), while the agreement weakens for the coral-only reconstruction (Fig. 10d,





middle column).  This suggests the Fogt and Connolly (2021) dataset aligns more with tropical
variability in the early 20[th] century, which dominates its response, but more with ice core related
variability in the latter half of the 20[th] century, which dominates the paleo-based reconstructions
near Antarctica.  Further, given the greater skill in the O'Connor et al. (2021, 2023)
reconstructions relative to Dalaiden et al. (2021) in the later period, this also suggest that
including external forcing in the climate model prior is important.

Since there was some effect of seasonality on the relationship between the paleo-based and

statistically based sea ice reconstructions in Fig. 4, we examine the correlations for each season
(like with the Fogt et al. (2022a) sea ice reconstructions, the Fogt and Connolly (2021) pressure
datasets were constructed seasonally, and annually averaged for Fig. 10, Table 1) in the 1905-
1956 period in Fig. 11.  However, for the pressure datasets, the change in seasonal skill in Fig.
11 is dampened compared to that for Antarctic sea ice extent in Fig. 4.  There are more
correlations statistically significant ($p<0.05$) for the Dalaiden et al. (2021) dataset in DJF and
SON, likely tied to when the skill of the Fogt and Connolly (2021) dataset is the highest.  For the
O'Connor et al. (2021,2023) datasets (Figs. 11b-d), the agreement is slightly better over the Ross
Sea and West Antarctica in winter.  This improvement is due to contributions from both ice cores
and corals, with the corals providing enhanced agreement off the coast and across East
Antarctica (Figs. 10c,d).  The ice core-only reconstruction provides better agreement over West
Antarctica in SON (Fig. 10d, right column).  Another interesting feature, worthy of further study,
is the opposite agreement in the Weddell Sea in MAM between the ice core-only reconstruction
(Fig. 10c, significantly [$p<0.05$] positively correlated) and the coral-only reconstruction (Fig.
10d, significantly [$p<0.05$] negatively correlated); the significant positive correlations over the
Weddell Sea onto the Antarctic continent in MAM are also seen in the Dalaiden et al. (2021)





reconstruction, related to the ice core constraints (since this dataset does not assimilate coral
proxies).

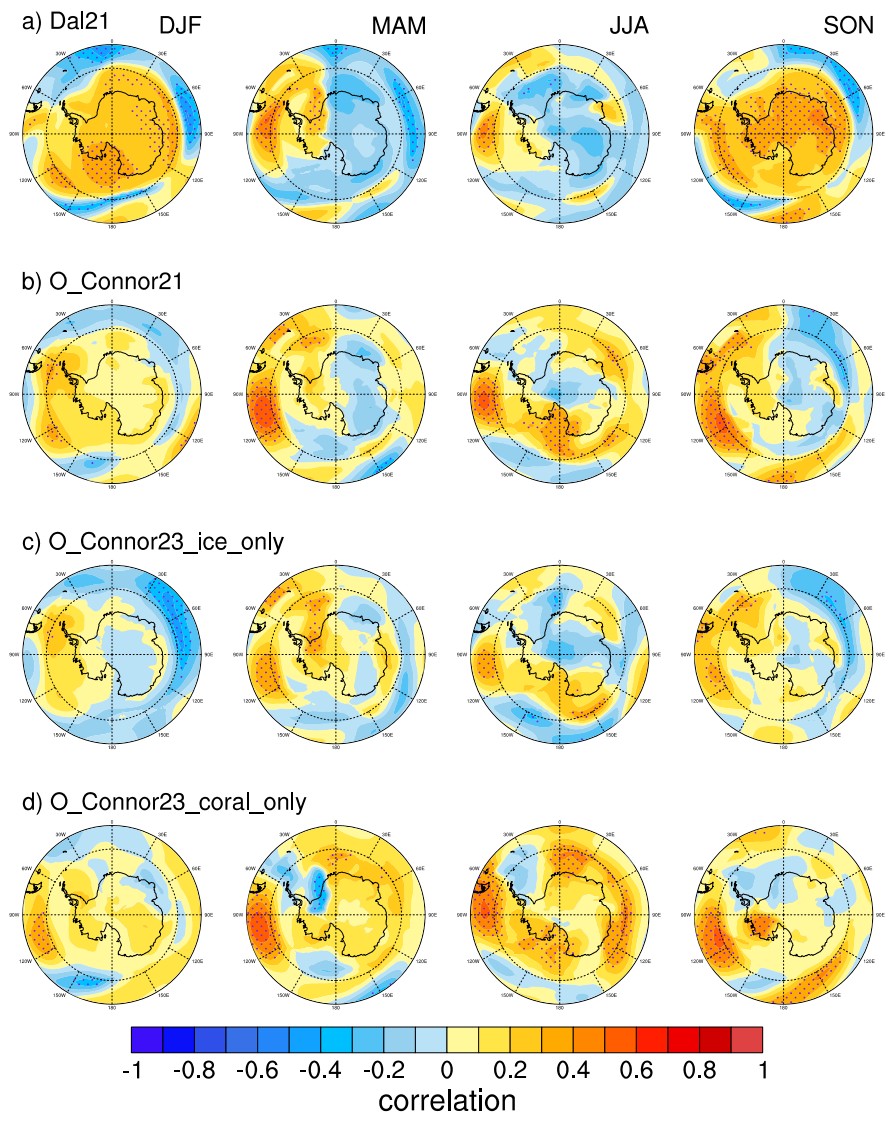

**Figure 11.** As in Fig. 10, but for correlations of the annual mean pressure reconstructions with
the seasonal mean reconstruction of Fogt and Connolly (2021) during 1905-1956. The columns,
from left to right, represent DJF, MAM, JJA, and SON, respectively.





To more completely evaluate these datasets, annual mean comparisons with select
observations are conducted in Fig. 12 using the gridpoints nearest to the station locations (for
reference, a map of the stations is provided in the legend of Fig. 12). In the full period 1905-
2000 (left column), correlations of the paleo-based datasets with the Fogt and Connolly (2021)
dataset are generally near zero, and only significant (here, $p<0.01$) over the West Antarctic
continent at Byrd station. The correlations with the observations improve after 1945 (right
column in Fig. 11), most now significant at $p<0.01$, especially for the O'Connor et al. (2021)
dataset. The inclusion of more proxy data in the O'Connor et al. (2021) dataset (Fig. 2c) and the
inclusion of an anthropogenically forced prior (Table 1) likely lead to overall better agreement
with the observations than for the Dalaiden et al. (2021) dataset; however both of these paleo-
based datasets use a moderately coarse spatial resolution prior that may limit their comparison at
a single point. It is also important to note that while the Fogt and Connolly (2021) correlations
are overall higher than the paleo-based, this dataset is calibrated directly to these observations,
and for Orcadas, direct observations (and not a reconstruction) are used, hence why the
correlations are strong in both time periods at this location, black correlation values at top of the
figures in the bottom row of Fig. 12).
More striking than the change in correlation in Fig. 12 are the changes in the linear trends
through time across the various datasets (given by the values at the bottom of each panel). In the
full time period (1905-2000, left column of Fig. 12), both the Dalaiden et al. (2021) and
O'Connor et al. (2021) datasets show statistically significant ($p<0.01$) pressure decreases at Byrd
and Halley stations (Figs. 12a,c), but statistically significant ($p<0.01$) pressure increases at
Orcadas (Fig. 12d). The Fogt and Connolly (2021) dataset, in strong contrast, shows significant
($p<0.05$) pressure increases at Byrd in both time periods (consistent with observations after 1957,

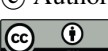

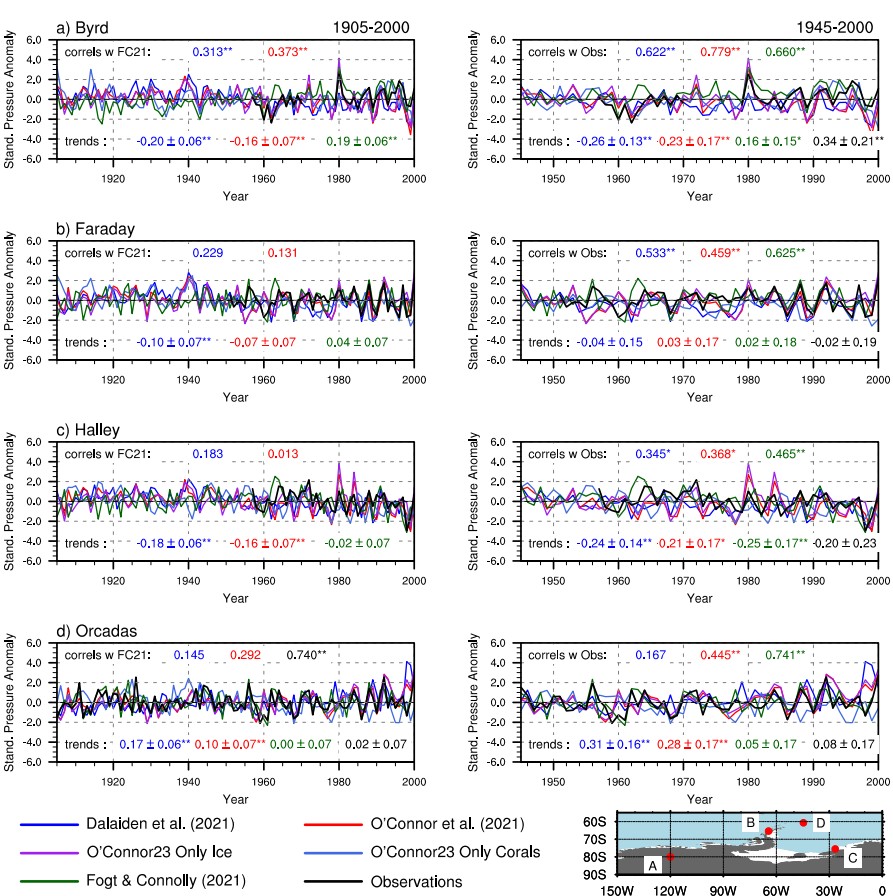

**Figure 12.** Timeseries of standardized pressure anomalies at various locations given in the map at the bottom right. The observations, mostly starting around 1957 (1903 for Orcadas, panel d), are shown in black. For the left column, correlations given at the top of each panel are with the Fogt and Connolly (2021) dataset during 1905-2020, and trends at the bottom are calculated for the 1905-2020 period. For the right column, the correlations are with the observations, and both correlations and trends are calculated during 1945-2020. In all panels, correlations and trends significantly different from zero at $p<0.05$ and $p<0.01$ are marked with * and **, respectively.

Fig. 12a), and only significant ($p<0.01$) pressure decreases at Halley station after 1945 (Fig. 12c,

right column). The much stronger trends in the paleo-based pressure reconstructions compared

to the Fogt and Connolly (2021) dataset during 1905-2020, and even with most observations



after 1945, are undoubtedly a strong contributor to the differences seen in the Antarctic sea ice
extent reconstructions between the paleo-based and statistically based reconstructions; there are
statistically significant differences in the underlying atmospheric circulation used to generate
these reconstructions that connect to their differences in trends seen in Figs. 3 and 6.

Both the Dalaiden et al. (2021) and O'Connor et al. (2021) studies focus on marked pressure

decreases in the region of the Amundsen Sea low, which as discussed earlier, has a strong role in
regional sea ice conditions examined in this study (Hosking et al., 2013; Raphael et al., 2016;
Holland and Kwok, 2012).  It is much more challenging to investigate the reality of these long
term changes in this region, as there are no direct observations (the closest stations are
represented by those in Fig. 12), and the Fogt and Connolly (2021) dataset shows the lowest skill
(compared to the ERA-Interim reanalysis) spatially in this region, consistent with the fact that
even contemporary reanalyses show the greatest disparity (in terms of pressure correlations and
trends) in this region as well (Fogt et al., 2018).  Given the disparities in trends seen from
observation locations in Fig. 12, it is not surprising that the datasets also have dramatically
different trends in this region, which have similar implications for differences in sea ice extent
trends. To visualize this in perspective, a full spatial comparison across the Southern Hemisphere
of the pressure trends in the various datasets in three time periods, along with available
observations, is provided in Fig. 13 as a final comparison of these products.

First, for the period of Antarctic observations (1957-2000, middle column of Fig. 13),

observations indicate statistically significant (boxes, $p<0.05$) pressure decreases only over
coastal East Antarctica, with insignificant positive annual mean trends over the East Antarctic
plateau at Vostok (only observation plotted in central East Antarctica in Fig. 13, consistent with
ERA5 surface pressure trends during 1979-2022 in Fig. 1b).  At Byrd station in central West

Spatial Trends with Observations

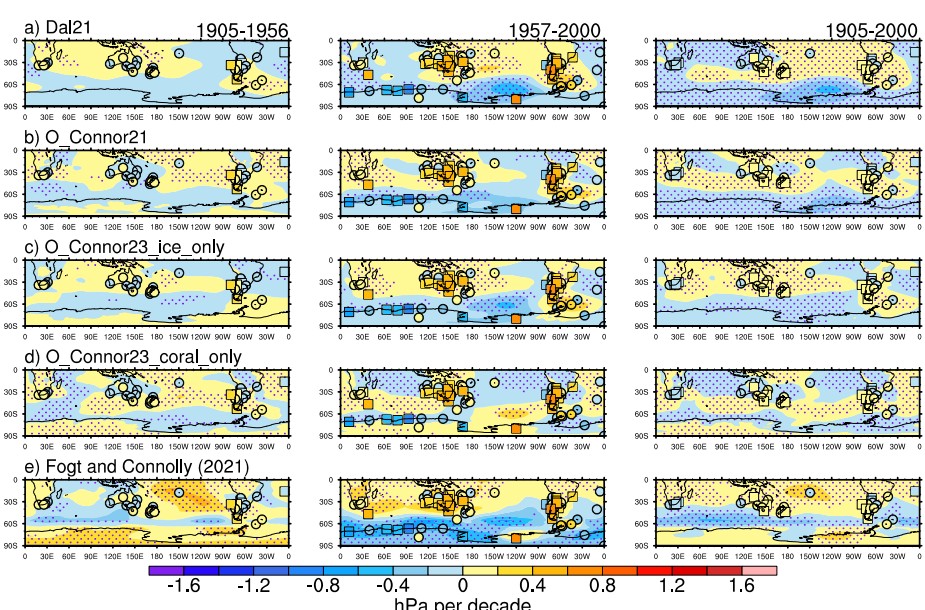

**Figure 13.** Linear near-surface annual mean pressure trends (hPa per decade) for 1905-1956 (left column), 1957-2000 (middle column), and 1905-2000 (right column) for various datasets in each row a) Dalaiden et al. (2021); b) O'Connor et al. (2021) proxy reconstruction c) O'Connor et al. (2023) reconstruction with only ice core proxies; d) O'Connor et al. (2023) reconstruction with only coral proxies; e) Fogt and Connolly (2021) station-based reconstruction. Trends from observation data, where available, are plotted using the same color scale on top of the gridded pressure trends. Stippling indicates gridded pressure trends significantly different from zero at $p<0.05$, boxes indicate trends in observations significantly different from zero at $p<0.05$.

Antarctica, statistically significant positive trends (annual mean) are observed (also seen in Fig.

12a). The various datasets capture these patterns to differing degrees: paleo-based datasets

generally only have statistically significant negative trends along the Antarctic coastline,

consistent with the positive annual mean SAM trend (Fogt and Marshall, 2020), but only capture

positive trends in the coral-only reconstruction (middle column Figs. 13a-d). Meanwhile, the

Fogt and Connolly (2021) dataset incorrectly has significant ($p<0.05$) negative pressure trends

over the East Antarctic plateau, but does capture a regional pressure increase in West Antarctica



(Figs. 13e middle column, Fig. 12).  The Dalaiden et al. (2021) dataset has a marked significant
deepening of the Amundsen Sea low during 1957-2000 (Fig. 13a, middle column), which is
largely driven by ice core paleo data (Fig. 2b), and thus reduced in the O'Connor et al. (2021)
dataset by the inclusion of coral data (Figs. 13b-d; Fig. 2c).  This suggests that if the significant
pressure increases over West Antarctica are correct, there is likely a tropical signal to these
positive trends (to be discussed in more detail later).  In the midlatitudes during 1957-2000, the
coral-only reconstruction (Fig. 13d) captures the pattern seen in observations the best (the Fogt
and Connolly (2021) dataset is not independent of observations in these locations since it is
merely the 20$^{th}$ century reanalysis north of 60°S, Table 1).  It should be noted that there are large
inconsistencies with observations at Tahiti in the central tropical Pacific and the 20CRv3, as
discussed in Fogt and Connolly (2021).

For the early 20$^{th}$ century during 1905-1956, the clear difference in trends in 1905-1956

between the Fogt and Connolly (2021) dataset and those of Dalaiden et al. (2021) and O'Connor
et al (2021) is distinct.  Pressure trends over the Antarctic continent reverse in the Fogt and
Connolly (2021) dataset, while they remain insignificant in most paleo-based datasets.  It is
interesting however that the coral-only reconstruction from O'Connor et al. (2023) (Fig. 13d)
also produces statistically significant positive trends over Antarctica in all time periods,
consistent with the Fogt and Connolly (2021) dataset during 1905-1956, and aligning with the
period when these two datasets agree the most (Figs. 11-12). Notably, this coral-only dataset
agrees the best with observations in the midlatitudes during 1905-1956 (Fig. 13d, left column),
and is quite similar to the pattern in the early 20$^{th}$ century from the full dataset in the midlatitudes
(Fig. 13b, left column), suggesting a primary role of the coral records in constraining the solution
in the midlatitudes.  The larger sample size afforded by the full time period from 1905-2000





yields more statistically significant trends across the Southern Hemisphere in the paleo-based
datasets (Figs. 13a-d, right column) that are overall similar in structure to the trends in the paleo-
based datasets during 1957-2000 Figs. 13a-d, middle column).  The coral-only dataset slightly
produces better agreement with the midlatitude observations, capturing the significant negative
trends over South Africa that are also hinted at by insignificant negative pressure trends at Perth
in southwestern Australia. In comparing the paleo-based with the station-based Fogt and
Connolly (2021) datasets, the biggest differences are the trends poleward of 60°S, which are
insignificant except for pressure increases (p<0.05) over West Antarctica in the Fogt and
Connolly (2021) dataset due to the cancellation of different trends in the early and late 20$^{th}$
century in this dataset (Fig. 13e).  Compared to the other sources of differences in the Antarctic
sea ice reconstructions examined earlier, this connection to the implied changes (or lack thereof)
in the atmospheric circulation in the 20$^{th}$ century are a dominant contributing factor to the
differences in the linear trends among the Antarctic sea ice estimates seen in Fig. 2; this similar
conclusion was reached in Fogt et al. (2022b) using a much more limited analysis.

**4.  Discussion**

While this study largely focuses on causes for the differences between various sea ice and

pressure reconstructions, it is important to note that there are broad similarities as well,
especially in the Weddell sector where cross-correlations are strongest, even stronger in some
seasons than in the annual mean.  Further, each of these datasets has their own limitations due to
their methodology and assumptions made in their reconstruction procedure, and that every
reconstruction is only an estimate and is not without error.  Nonetheless, the differences in the
sea ice extent reconstruction trends (and to a lesser degree, the interannual variability) largely





appear to hinge on whether or not there was a change in the atmospheric circulation in the high
southern latitudes (poleward of 60°S) in the 20th century.  Paleo-based reconstructions,
constrained by ice core records over Antarctica, have similar pressure trends through time, which
influence the proxy-based sea ice extent reconstructions.  In contrast, station-based
reconstructions of both pressure and Antarctic sea ice extent show trends that change through
time, which reduces their similarities with proxy-based reconstructions (for both sea ice extent
and pressure) in the early 20th century.  Given that coral-only reconstructions from O'Connor et
al. (2023) agree better with the Fogt and Connolly (2021) dataset in the early 20th century, there
it is possible that if there was indeed a reversal of atmospheric circulation trends, that these had a
tropical association to them.  Nonetheless, the analysis here shows an important influence of the
implied atmospheric circulation on Antarctic sea ice variations in both proxy-based and station-
based sea ice extent reconstructions, consistent with findings from observations and models (Sun
and Eisenman, 2021; Blanchard-Wrigglesworth et al., 2021).
In looking more closely at the analysis presented here, the limited observational data always
present a challenge.  In particular, in the vicinity of the Amundsen Sea low, Byrd station in West
Antarctica (see map in Fig. 11) is the only observational record spanning over 30 years in the
150°-90°W sector poleward of 60°S (not including the South Pole).  While weather observations
started at this station in 1957, it shut down for a period in the 1970s, and was replaced by an
automatic weather station nearby.  While there has been considerable work done to patch the
temperature record at this station (Bromwich et al., 2012), there could be measurement-specific
errors that lead to the positive trend at this station in the observational dataset that may at least
partially explain the regional differences in pressure (across West Antarctica and extending
northward into the Amundsen and Bellingshausen Seas) between the paleo-based pressure



datasets and those of Fogt and Connolly (2021). However, the AWS in the later portion of the
record is at a slightly higher elevation than the station observations in the earlier part of the data,
which would generate negative pressure trends based on instrument elevation changes, opposite
the positive annual mean trends seen here. There is also a strong seasonal pattern to the positive
pressure trends over West Antarctica, as shown in Fogt et al. (2018). Further work is needed to
investigate the Byrd pressure data source, as well to employ historical data from ship logbooks
and early expeditions (Edinburgh and Day, 2016; De La Mare, 2009; Fogt et al., 2020; Lorrey et
al., 2022) to better understand possible atmospheric circulation shifts in the high southern
latitudes prior to 1957. Additionally, each kind of observations (e.g., paleo or instrumental
observations) has its own advantages and weaknesses. Combining these two sources of
information could provide a more accurate reconstruction of historical surface climate changes.
**5. Conclusions**

The analysis presented in this paper has evaluated various sea ice extent reconstructions

spanning the Ross Sea eastward to the Weddell sea, and pressure reconstructions across the
entire Southern Hemisphere to explain the differences between the sea ice extent reconstructions.
Overall, better agreement in the sea ice extent reconstructions was found in the Weddell sea,
despite possible changes in the relationship sea ice shares with the atmospheric pressure in this
region throughout the 20$^{th}$ century. In the Ross and Bellingshausen seas, the agreement is
weaker, and appears to be more strongly tied to the atmospheric circulation. For ice core based
reconstructions studied here, the Thomas and Abram (2016) reconstruction is fairly consistent
with other reconstructions in the Ross sea, while the differing spatial and temporal representation
of the Abram et al. (2010) reconstruction make it challenging to effectively compare to other
datasets. Overall, paleo-based and station-based pressure reconstructions give notably different



trends from the Ross Sea east to the Weddell Sea throughout the 20[th] century, especially in the
vicinity of the Amundsen Sea low, a semi-permanent pressure cell known to strongly modulate
Antarctic sea ice in the Ross, Amundsen, and Bellingshausen Seas (Hosking et al., 2013;
Raphael et al., 2016).

While this study has determined that many of the differences in the paleo-based and

statistically generated Antarctic sea ice extent reconstructions are tied primarily to their
connection to the underlying atmospheric circulation, it is not able to determine the validity of
changes in the atmospheric circulation in the early to mid 20[th] century.  Further data extraction
from ship logbooks (Lorrey et al., 2022), new climate model simulations assimilating both paleo
data and observations, and isolated forcing simulations in coupled climate model simulations
may help to understand this potential atmospheric circulation change.  Nonetheless, this study
has helped to better understand the differences and relative strengths and limitations of not only
the Antarctic sea ice reconstructions examined here, but also several pressure reconstructions.  It
is thus hoped that future users of these valuable datasets will exercise the necessary caution when
analyzing them given the knowledge of the processes they represent well, and other mechanisms
that they may not reproduce as reliably.
**Code and Data Availability**
The Fogt et al. (2022a) sea ice reconstructions can be obtained from the National Snow and Ice
Data Ceter dataset G10039 (https://nsidc.org/data/g10039/versions/1) (Fogt et al., 2023).  The
Fogt et al. (2019) and Fogt and Connolly (2021) pressure reconstructions can be downloaded
from figshare (https://doi.org/10.6084/m9.figshare.c.6765447.v1). Data for the Dalaiden et al.
(2021) sea ice extent and pressure reconstructions are available on Zenodo
(https://zenodo.org/record/4770179).  The O'Connor et al. (2021) pressure reconstructions are



available on Zenodo (https://zenodo.org/record/5507607#.Y6OOl-yIb0o) and the sesnsitivity
reconstructions from O'Connor et al. (2023) can also be downloaded from Zenodo
(https://zenodo.org/record/8007655).

**Author Contributions**
RLF and QD designed the study. All authors contributed to writing and editing the manuscript,
and RLF produced the figures.

**Competing Interests**
The authors declare no competing interests.

**Acknowledgments**
RLF acknowledges support from the U.S. National Science Foundation Office of Polar Programs
award #1744998. QD is a Research Fellow within the F.R.S.-FNRS (Belgium).



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
