# Peer review of "A Comparison of South Pacific Antarctic Sea Ice and Atmospheric Circulation Reconstructions Since 1900 Ryan L. Fogt1, Quentin Dalaiden2, and Gemma K. O'Connor3 1Department of Geography and Scalia Laboratory for Atmospheric Analysis, Ohio University, At"

_Climate of the Past, 2023_

## Referee Comment (RC1)

**SUMMARY: GENERAL COMMENTS**

In light of recent extremes in Antarctic sea ice, and debate over their climatological significance and possible causes, this is a very timely, important, and thorough paper on interpretations of longer-term sea ice reconstructions. My take homes are, first, that sea ice reconstructions differ a lot pre-satellite era, and this is only partially explained by season or region being represented. Second, there is evidence that different implicit underlying atmospheric reconstructions drive these changes, in particular changes in the sea ice-pressure relationship over time and indeed in pressure trends themselves over time. There are also lots of other analysis details and interesting corollaries that will be of interest to the community. It is certainly relevant for publication, and I have no concerns with the analysis or methodology.

However, I do have major comments on interpretation and presentation in a broad sense; the text becomes overly detailed and at times the scientific reasoning that leads to conclusions is hard to follow and therefore unconvincing. Many methodological details are unnecessarily repeated in the results section, which makes it hard to follow the results; many data details are missing. The conclusion section is vague and could benefit from strengthening with some specific take-home messages. Figures 3, 4 and 12 also require reworking by the addition of correlation tables and other edits. I've given fairly detailed suggestions for all these points under 'specific comments'.

I therefore recommend this manuscript for publication after minor revisions; the revisions I suggest are large in number but would not, in my opinion, require a second round of review.

**SPECIFIC COMMENTS**

L26: 'has with its underlying' unsatisfactory: the proxy-based records don't explicitly have an underlying atmospheric circulation?

L29: paleo-based-> proxy-based?

L31 -> "sensitivity experiments …" This sentence is very hard to understand without having read the paper multiple times! I suggest simplifying what you try to convey: something like 'results from reconstructions based only on coral or ice core records, rather than both, imply contrasting roles of these records- and therefore of tropical versus purely local atmospheric variability- in driving different types of reconstructions' or similar.

L53: For NSIDC Sea Ice Index SIE, 2022 also a record low annual-mean, so include here.

Last line of the abstract emphasises tropical versus local drivers as a key take home but this isn't mentioned in conclusions and for me wasn't a main focus of the paper. Either change this line of the abstract or add a short discussion of this to the conclusions section!

**Introduction:** Some of the discussion in introduction seems to be about general Antarctic climate (e.g. line 66 'across the continent') not sea sea ice. Clarify/refine text? E.g. line 69 'Antarctica' ->'Antarctic sea ice and atmospheric variables'; similarly line 71 'climate' -> '? Temperature? Sea ice?'

L62 'near 1978-1979' -> 'in late 1978'

**L65-67:** Mention predicting future change as a motivation for understanding the past

**L77:** Add a comment on time coverage of these other instrumental observations outside Antarctica, e.g. 'dating back to as far as …'

L100 Clarify the scope of the study: mostly concerned with West Antarctic sea ice, annual focus but seasonal is examined to help understand annual.

**L116-118** it took me a lot of mental effort to reconcile this with the table. Try 'We also investigate two proxy-based reconstructions of spatially limited sea ice extent, and three spatially complete proxy-based climate model reconstructions, two of SLP and one of sea ice.'

**Section 2.1).** Ensure each dataset (sea ice obs, sea ice recon, pressure recon) has its own paragraph- the details become blurred. A couple of lines are needed introducing the Fogt et al 2022 sea ice reconstruction. Is the best fit reconstruction used here? How is the annual mean created (move here from results).

L154: 'Equatorward extent of the' -> 'total' (equatorward extent would imply e.g. polynya area was counted as extent?)

**L211:** Information about observations (Met obs for Fig 12, Met obs for Fig 13, and Weddell fast ice) should be included here and any repetition removed from results. Detail should be added on the time coverage of station data in Fig 12, and how these stations were selected.

**L175 and Fig 1:** The black lines on Fig 1 (specifically 70-100W) are not the same as sectors in Parkinson (2019). I don't think Parkinson sectors are needed- when Raphael and Hobbs sectors are introduced it should just be mentioned that they differ from traditional sectors (with ref to their paper which describes it fully). Fig 1 should just indicate 100-70W for the Abram reconstruction. Also, decrease size of green dot slightly for more precise rendering of location.

**L182-210**: Explain clearly the 'fixed prior' versus forced prior that's discussed later in the section and in Results, and state which model uses what. Add this info, and whether the data is directly calibrated against obs, to Table 1. L191 'temporal variability comes only from proxies' tropical pacemaker/external forcing prior surely forces temporal variability?

**L217-218** 'standardised (to place on same scale) annual-mean sea ice extent' -> 'standardized annual-mean sea ice extent anomalies'. Recap why magnitudes differ (because of region definitions) and variability not magnitude matters.

**L224-228:** 'Highest' implies multiple datasets being compared but at any point in this sentence there are only two (Dalaiden versus EITHER Thomas and Abram OR Abram ). A different emphasis is therefore needed e.g. '[T&A] correlations with observations exceed those of Dalaiden 2021 in Ross-Amundsen, but Dalaiden exceeds Abram 2010 in both Bell-Amundsen and in Weddell'

**L226 etc:** 'data assimilation based' etc. gets repetitive explaining what reconstructions are every time (e.g. also line 245-246, 302,340,…) and the resultant wordiness detracts from scientific reasoning. I suggest introducing a shorthand e.g. Paleo=>'PALEO reconstruction', station based=>'STAT reconstruction', estimate from data-based reconstruction=> 'ASSIM reconstruction' and abbreviation for dataset names. E.g. L226 would become something like 'while the Dal_21 ASSIM reconstruction …'

L324 'Sectors have been adjusted' i.e. the SIE has been recalculated from Dalaiden SIC using new boundaries? Reword for clarity.

**Fig 3, 4:** These are very hard to decipher, not least because red is different dataset in different panels. Specify this in the caption. Please put the correlations in cross-correlation tables as a new right hand column to this figure. At the moment working out which correlation is which, reconciling with the text, and in particular comparing correlation values, is extremely challenging; a table would

greatly improve this. (Also, the correlation text at bottom of Fig 3 panels uses different layout to those of Fig 4 panels- very confusing.)

**L242** Context needed: where correlations with obs 1979-present exceed cross-reconstruction correlations 1905-2020, how do cross-correlations compare over the common 1979-2020 period?

**L242-259** It is hard to follow this discussion of comparison of correlations, partly due to the paragraph structure and partly due to the lack of correlation tables. Indeed I think it is misleading as the text seems to state cross-correlations are all lower than correlations with observations, then later states this isn't true for Abram 2010. To resolve, I'd use a table and address correlations in turn: first Dalaiden with pure proxy-based then Fogt with all other datasets. Also, be sure to sell the take home here; basically none of the cross-correlations are significant pre-1979 so the reconstructions really do tell us very different things!

**L260** 'Sudden anomalies': very intriguing in light of current narrative around records. Suggest expanding this, or commenting on implications (e.g. extremes from paleo record should be interpreted with great care- see also end of my previous comment?!)

L273: To aid with seasonal vs annual, compare Fogt SON to Abram et al (2010) ASO?

L276 'biased by accumulation at the ice core site': cite?

L286: r=0.56 isn't so different from r=0.525, similarly 0.345 vs 0.320! I'd drop line 285-289 to avoid losing the woods for the trees, and slightly rephrase L290 accordingly. Similarly line 305-307 doesn't seem to describe something that's substantially different from the annual? Is line 310-311 (better agreement in seasonal) therefore true?

L304 I have a general dislike of calling 'SON' spring for sea ice- it's the maximum! Meterologically correct, cryospherically misleading. Rephrase.

Figure 5: Indicate the relevant sector in each panel, to draw the readers eye to where one might expect the correlations to be high and positive.

L382 remove bracketed phrase: repetition

Fig 6: The satellite trends look like they start in 1969?

**L398:** this paragraph is crucial but a bit weak. At L399 (and 414) expansion/recap is needed on why Fig 5 implies role of atmospheric circulation? For me it's more Fig 6 (contrasting trends which we know in obs period are linked to wind patterns) that imply the circulation role? At line 406, Dalaiden pressure is linked to Fogt sea ice, but this seems a bit of a leap of reasoning?

**L409:** Very helpful paragraph!

**L419:** Split this section for readability. 3.2 'Connection to the atmospheric circulation changes' and add at L529 3.3) 'Differences in atmospheric reconstructions'

**L424** 'for consistency' with what?

**L424-430** over-repetition of methods. Cut down and reference Methods section

**L430-433.** "Since the Antarctic station pressure reconstructions were generated using a similar statistical technique as the Fogt et al. (2022a) sea ice extent reconstructions, this allows for an evaluation of other estimates that are expected to provide **similar temporal variability** as the Fogt et al. (2022a) sea ice extent reconstructions." Seems at odds with the conclusions that the

relationship between Fogt sea ice and Fogt pressure changes over time and with statements at L466-469. I think this is just me being confused, so please elaborate for clarity of reasoning!

*L434:* "Figure 7 displays the correlations for the Weddell sector sea ice extent from Dalaiden et al. (2021) with the gridded pressure datasets in the top rows, and the Weddell sector sea ice extent reconstructions from Fogt et al. (2022a) with the same gridded datasets in the bottom rows" -> "Figure 7 displays the correlations for the Weddell sector sea ice extent from (top rows) Dalaiden et al, 2021) and (bottom rows) Fogt et al. (2022a) with three gridded pressure datasets"

**L439** 'panels' -> 'rows'?

**L456-L461:** (overly lengthy) Change to : "However, **for** the Fogt and Connolly (2021) pressure dataset, the relationship **with** Weddell sea ice extent shows a change in **sign** poleward of 60°S in the 1905-1978 period".

L471: 'this pattern' what pattern?

L503: clarify Orcadas is met data not sea ice.

L511- 'change sign' but neither sea ice time series has significant correlation with Orcadas SLP so I think not much can be read into this.

L512: 'spatial pressure change' -> 'change in the Weddell SIE-gridded SLP correlation change'

**L516 and other locations**: add explanation of why analysis is split at 1945.

L517 'weakly correlated' –> 'have no statistically significant correlations'

L538-552: (presentation) Overly lengthy, repeats content from methods, and implies slightly that new sensitivity runs were performed for current paper. Suggested succinct reword: 'Further, we analyse the single-proxy-based reconstructions of O'Connor (2023) to pinpoint the sensitivity of these correlations to using only tropical (coral) or Antarctic (ice core) paleo constraints'. Also this is mostly a repetition of methods section.

L552 New paragraph. Re-explain why 1956 break point?

L552-L602: This section is hard to follow and needs more careful wording/separation of 1) discussion of what's driving the patterns in the reconstructions (comparing panel patterns), 2) how well they are correlated with Fogt and Connolly and 3) what we can learn (i.e. what's driving variability in Fogt and Connolly? Or is Fogt and Connolly considered as more reliable?) Is one possible interpretation that Fogt and Connolly is biased in East Antarctica due to over-dominance of tropical here in this reconstruction? Specifically the word 'agreement' is ambiguous e.g. at line 569 'agreement improves' –agreement between assimilation reconstructions, or with Fogt and Connolly?

L556: I think this implies O'Connor 2021 has forced prior but that L456 implies it has fixed?

L563-564: rephrase to 'not surprising that the coral-only dataset of O'Connor et al is better correlated with F&C'- but this is only true pre-1957?

L566- 'is opposite that of… especially in East Antarctica' I'd rephrase as 'except in the Amundsen-Bellingshausen sector' and then include the note about teleconnections being less important for East Antarctica.

L581-L582: belongs in methods

L592 'agreement' -> 'correlation'

L619: 'changes' (implicitly in time) or 'differences' (between datasets)?

Fig 12: Again consider correlation tables. Which value refers to which dataset is very hard to interpret. The Fogt dark green and obs black are almost indistinguishable. What's the LHS? Text (L608) 'correlations with obs improve after 1945' implies it is correlation with obs but text in panels and the fact most obs aren't available until 1945 implies not. Also L619 "change in correlation in Fig 12" what's meant here?

L641 'statistically significant differences in the underlying atmospheric circulation' are there?

L660: Give Vostok longitude.

L678: The word 'incorrectly' is important; it implies Fogt and Connolly early C20 trends are in fact incorrect, which sheds information on which of all the reconsturctions is more reliable. Discuss.

L693: 'reverse' when? Previous sentence is only discussing pre-1956 but trend reversals are later?

Discussion: Byrd data limitations should be mentioned briefly in results section.

L778: "not able to determine the validity of changes in the atmospheric circulation in the early to mid 20th century." … and therefore not able to deduce which sea ice reconstructions are the best representation of reality?

L782-> (on strengths and weaknesses and lessons for users): I wonder if this could be recap some key specific conclusions from the paper? Or if the authors feel uncomfortable doing this, flag the key questions that remain to be answered to understand which datasets are more reliable.

**Technical Comments**
L18 minima -> lows
L20 add 'since 1900'
L27 'several'-> '5'
L113 remove 'various'. Put this sentence to end of para.
L125 link failed but https://rda.ucar.edu/datasets/ds570-0/ worked
L151 'combines of' -> 'combines'
Table 1 column 2 heading 'type' sufficient. In column 4 also ref Figs 2b) and c) where relevant
Table 1 last row/column 'interpolated' typo
L174 Remove 'a' before 'three'
L219 with->of
L224 ', and the Thomas and Abram…' -> '. The Thomas and Abram…'
L242: Remove 'however'
L246 '; part'-> '. Part'
L264 (end): 'created in'->'created with'
L279 'Compared to' -> 'in contrast with'
L428 'A blend interpolated' -> 'a blend of interpolated'
L553: 'between' -> 'for' (authors are not cross-correlating these datasets with each other)
Fg 12 Be explicit that point A = panel a)
L689 'with' -> 'between'
L691 repeated '1905-1956'
**L791 'Ceter'->'Center'**

---

## Referee Comment (RC2)

**A Comparison of South Pacific Antarctic Sea Ice and Atmospheric Reconstructions Since 1900**

The study compares different proxy-based and station-based Antarctic sea ice extent reconstructions throughout the 20th century with the aim to investigate the reason behind their discrepancies, with a particular focus on atmospheric circulation. The reconstructions show a good agreement in the Weddell sea, and a weaker agreement in the Ross and Bellinghausen seas possibly due to changes in atmospheric circulation. The sea ice reconstructions are compared with pressure reconstructions spanning the Southern Hemisphere, finding a connection of mid-latitude atmospheric circulation with Antarctic sea ice variability.

The study is very well detailed and definitely worth publication. I particularly appreciated that years of efforts in reconstructing sea ice using paleoclimate archives are being used in a wide spatial and temporal context to investigate underlying climate relationships. There are a few typos here and there but otherwise it is excellently written, with a good flow and level of English. However, while I appreciate the level of detail of the paper, I think it could have been more concise, in order to facilitate the reader to go through the very long text. I suggest to try to simplify the text whenever possible as I had myself to read the paper several times to understand the reasoning of the authors. One possibility could also be to report a discussion of the results in the "Discussion" section that follows the same structure as the "Results" section, to better synthesize the main findings of the article.

Also, as the authors state on page 15, the various sea ice reconstructions represent different aspects of sea ice variability and moreover are based on different methodologies and on a different number of records considered. I suggest to be more clear of these limitations in the text when examining the different sea ice reconstructions and their relationship with atmospheric circulation.

Please find below some technical comments:

Table 1: in the Abram et al. (2010) an Thomas and Abram (2016) rows I would indicate the type of sea ice proxy used. In the Dalaiden et al. (2021) row please modify "isotope-enable CESM1" with "isotope-enabled CESM1"

Figure 2a: please change the title to Fogt et al. (2022a) for consistency

Line 151: "The CDS algorithm output combines ice concentration". Remove "of".

Line 169: "sediment-derived"

Line 172: "ice core-derived"

Line 174: remove "a" from "used a three ice cores"

Line 176: "Further West" with capital letter

Line 190: "and those variables given by the Earth System Model…" please specify which variables

Line 194: the Dalaiden et al. (2021) reference is repeated twice, please remove one

Line 216: please change with "paleo data, including those from data assimilation -based reconstructions"

Figure 4: please make clearer that the middle and right columns refer to Abram et al. (2010). Why not change the colors from Thomas and Abram (2016)?

Line 306: "Dalaiden et al (2021) are typically highest in JJA". Please remove "in"

Figure 5: I suggest to indicate the sectors in the figure as in figure 1a green line. Again, it is not clear which Fogt et al. (2022) paper is referred to.

Line 344-357: Here the authors compare the Abram et al. (2010) sea ice reconstruction with the other reconstructions in figure 5. However, such reconstructions are based only on three ice core records. I suggest to mention here that this is one of the limitations of that reconstructions with respect to e.g. Dalaiden et al. (2021) which is based on several records.

Line 368: typo in "geopotential height"

Line 391: "opposite to the strong positive trends". Please add "to".

Figure 7: as in Figure 2a please change the title to Fogt et al. (2022a) for consistency

Figure 8: Again clarify which Fogt et al. paper

Line 510: "it is not surprising that the relationship". Remove "then" and "both"

Figure 10: Please rename figure titles

Line 566: please add "to" after "opposite"

Line 573: "This suggests that the Fogt…". Please add "that"

Line 591, 593 and 594: please check the figure number, it should be figure 11

Line 608: Please refer to the figure

Line 616: I suggest to change "hence why the" with "explaining why the"

Line 617: add parenthesis before "black correlation"

Line 741: Figure 12 instead of 11

---

## Author Comment (AC1)

**We thank both reviewers for their comments and detailed feedback that have helped to improve the manuscript significantly. In response to both reviews, we have shortened the text as much as possible, and have used acronyms to refer to the various datasets we analyze in the paper. Although we have not added the tables suggested by reviewer 1, we have made it clearer in the text and figure captions that the correlations in each figure are color coded, which will hopefully clarify their values; we thought that adding other tables would not only increase the length of the paper, but also potentially add to the challenge of overly detailed and distracting text that was present in the first draft.**

**Our responses to the reviewer comments are in bold below.**

**Reviewer 1**
SUMMARY: GENERAL COMMENTS

In light of recent extremes in Antarctic sea ice, and debate over their climatological significance and possible causes, this is a very timely, important, and thorough paper on interpretations of longer- term sea ice reconstructions. My take homes are, first, that sea ice reconstructions differ a lot pre- satellite era, and this is only partially explained by season or region being represented. Second, there is evidence that different implicit underlying atmospheric reconstructions drive these changes, in particular changes in the sea ice-pressure relationship over time and indeed in pressure trends themselves over time. There are also lots of other analysis details and interesting corollaries that will be of interest to the community. It is certainly relevant for publication, and I have no concerns with the analysis or methodology. However, I do have major comments on interpretation and presentation in a broad sense; the text becomes overly detailed and at times the scientific reasoning that leads to conclusions is hard to follow and therefore unconvincing. Many methodological details are unnecessarily repeated in the results section, which makes it hard to follow the results; many data details are missing. The conclusion section is vague and could benefit from strengthening with some specific take-home messages. Figures 3, 4 and 12 also require reworking by the addition of correlation tables and other edits. I've given fairly detailed suggestions for all these points under 'specific comments'.

I therefore recommend this manuscript for publication after minor revisions; the revisions I suggest are large in number but would not, in my opinion, require a second round of review.

**We thank the reviewer for this incredibly thorough and helpful review – it has certainly aided us to improve the readability and flow of the text. This is one of the more detailed reviews I have encountered in my career, and we graciously thank you for your very careful read and the considerable time you invested in improving this manuscript!**

**We have decided to not add correlation tables as suggested, but have made it clear in the captions of Figs. 3,4, and 12 that the colors in each panel reference the various timeseries for which the statistics have been calculated. With this and the improvements (mainly shortening and focusing on key points), we hope that the discussion is now easier to follow around these figures without the need to add separate tables that would increase the length.**

L26: 'has with its underlying' unsatisfactory: the proxy-based records don't explicitly have an underlying atmospheric circulation?
**Changed 'underlying' to 'implied' - the proxy reconstructions still have an implied circulation connected with them that transports to the ice core.**

L29: paleo-based-> proxy-based?
**Changed as suggested**

L31 -> "sensitivity experiments ..." This sentence is very hard to understand without having read the paper multiple times! I suggest simplifying what you try to convey: something like 'results from reconstructions based only on coral or ice core records, rather than both, imply contrasting roles of these records- and therefore of tropical versus purely local atmospheric variability- in driving different types of reconstructions' or similar.
**Thanks for this suggestion. We have broken this into two sentences and reworded for clarity.**

Last line of the abstract emphasises tropical versus local drivers as a key take home but this isn't mentioned in conclusions and for me wasn't a main focus of the paper. Either change this line of the abstract or add a short discussion of this to the conclusions section!
**This was based on the comparisons with the Fogt and Connolly (2021) reconstructions with the coral only reconstructions from O'Connor et al. (2023), which generally agreed better in the earlier 20th century (Figs. 10, 11, 13). Have added a sentence to reflect this in the conclusions.**

L53: For NSIDC Sea Ice Index SIE, 2022 also a record low annual-mean, so include here.
**This was / is included, but separated from 2017 to allow for different citations specific to the 2022 record low.**

L62 'near 1978-1979' -> 'in late 1978'
**changed as suggested**

L65-67: Mention predicting future change as a motivation for understanding the past
**Done as suggested**

L77: Add a comment on time coverage of these other instrumental observations outside Antarctica, e.g. 'dating back to as far as ...'
**done as suggested**

L100 Clarify the scope of the study: mostly concerned with West Antarctic sea ice, annual focus but seasonal is examined to help understand annual.
**We have fully revised this sentence to shorten it and clarify the scope of the work.**

L116-118 it took me a lot of mental effort to reconcile this with the table. Try 'We also investigate two proxy-based reconstructions of spatially limited sea ice extent, and three spatially complete proxy-based climate model reconstructions, two of SLP and one of sea ice.' **We have shortened the sentence to remove the spatially complete text, as this is mentioned in the table and later in the analysis, so not necessary here.**

Section 2.1). Ensure each dataset (sea ice obs, sea ice recon, pressure recon) has its own paragraph- the details become blurred. A couple of lines are needed introducing the Fogt et al 2022 sea ice reconstruction. Is the best fit reconstruction used here? How is the annual mean created (move here from results). **We have separated each reconstruction and observations into their own section, and added a few more details here on the Fogt et al. (2022) sea ice extent reconstructions as suggested.**

L154: 'Equatorward extent of the' -> 'total' (equatorward extent would imply e.g. polynya area was counted as extent?) **Clarified that sea ice extent is just the area of grid cells with at least 15% concentration. It does in this way include area of polynya.**

L211: Information about observations (Met obs for Fig 12, Met obs for Fig 13, and Weddell fast ice) should be included here and any repetition removed from results. Detail should be added on the time coverage of station data in Fig 12, and how these stations were selected. **We have added these details here in the data/methods section as suggested.**

L175 and Fig 1: The black lines on Fig 1 (specifically 70-100W) are not the same as sectors in Parkinson (2019). I don't think Parkinson sectors are needed- when Raphael and Hobbs sectors are introduced it should just be mentioned that they differ from traditional sectors (with ref to their paper which describes it fully). Fig 1 should just indicate 100-70W for the Abram reconstruction. Also, decrease size of green dot slightly for more precise rendering of location. **Well spotted, the black lines were the Raphael and Hobbs (2014). We have removed the Parkinson (2019) boundaries and reduced the size of the green dot as well.**

L182-210: Explain clearly the 'fixed prior' versus forced prior that's discussed later in the section and in Results, and state which model uses what. Add this info, and whether the data is directly calibrated against obs, to Table 1. L191 'temporal variability comes only from proxies' tropical pacemaker/external forcing prior surely forces temporal variability?

**In data assimilation, the prior is the initial guess of the climate system that is updated using the available observations to obtain the best estimate of the state of the climate system, called the posterior. All the data assimilation-based reconstructions described in the manuscript are based on the same method to estimate the prior. Based on existing ESM simulations, the prior consists of the annual average of the state of the climate system (e.g., annual sea-level pressure, annual sea-ice extent). For instance, the prior used in the reconstruction of Dalaiden et al. (2021) consists of the 3003 years of the last millennium ensemble of iCESM1 (three simulations spanning the 850–1850 period). Through the data assimilation process, the prior does not vary over time, which means that the prior is identical for each year reconstructed, which we call a fixed prior. A forced prior is also fixed over time. The only difference comes from the simulations used for estimating the prior. In the case of the forced prior, the prior is estimated using simulations forced by anthropogenic forcing, like the pacemaker simulations of O'Connor et al. (2021, 2023; i.e., 1920–2005). In contrast, Dalaiden et al. (2021) excluded the historical period of the simulations to generate the prior. Therefore, in both cases, the temporal variability of the reconstruction only comes from the proxies. Although OCON21_ASSIM relies on a forced prior, the temporal variability of the forcing is not respected since the prior for reconstructing each year includes all the years of the simulation, from 1920 to 2005. We have specified in the updated version of the manuscript at lines xxx:**

**"It is worth mentioning that the prior used in the DAL21_ASSIM and OCON21_ASSIM reconstructions is fixed over time (i.e., each reconstructed year is based on the same prior) but while the prior used in the OCON21_ASSIM reconstruction includes the anthropogenic forcing, the reconstruction of DAL21_ASSIM solely relies on the natural climate variability as in Hakim et al. (2016) and Steiger et al. (2018). Therefore, since the prior remains constant over time, the temporal variability of the reconstruction only arises from the proxies."**

**and in Table 1.**

**Regarding the calibration of proxy records, in the DAL21_ASSIM reconstruction, ice core d18O and snow accumulation records are directly compared with the prior with no calibration against observation but for tree-ring width records, a calibration is done against local temperature and precipitation. In the case of OCON21_ASSIM, all the records are calibrated against local temperature except for tree-ring widths, which are also calibrated against precipitation. It is now specified in Table 1.**

L217-218 'standardised (to place on same scale) annual-mean sea ice extent' -> 'standardized annual-mean sea ice extent anomalies'. Recap why magnitudes differ (because of region definitions) and variability not magnitude matters.
**Added the word anomalies as suggested. However, we have not added the recap as the magnitudes can create different trends, which we focus on later, and there is worry that adding this makes the text even more wordy.**

L224-228: 'Highest' implies multiple datasets being compared but at any point in this sentence there are only two (Dalaiden versus EITHER Thomas and Abram OR Abram ). A different emphasis is therefore needed e.g. '[T&A] correlations with observations exceed those of Dalaiden 2021 in Ross- Amundsen, but Dalaiden exceeds Abram 2010 in both Bell-Amundsen and in Weddell'
**For brevity, we have deleted these sentences as they are small details and instead focus more on the differences between the paleo-based and station-based reconstructions.**

L226 etc: 'data assimilation based' etc. gets repetitive explaining what reconstructions are every time (e.g. also line 245-246, 302,340,...) and the resultant wordiness detracts from scientific reasoning. I suggest introducing a shorthand e.g. Paleo=>'PALEO reconstruction', station based=>'STAT reconstruction', estimate from data-based reconstruction=> 'ASSIM reconstruction' and abbreviation for dataset names. E.g. L226 would become something like 'while the Dal_21 ASSIM reconstruction ...'
**We have made these changes throughout (except in figures and figure captions, where readers may not fully read the paper and just skim these parts). We have also added a column in table 1 to mention the new acronyms, which we hope improves readability.**

L234 'Sectors have been adjusted' i.e. the SIE has been recalculated from Dalaiden SIC using new boundaries? Reword for clarity.
**Deleted this sentence and provided the details of this in the methods when this dataset was first introduced.**

Fig 3, 4: These are very hard to decipher, not least because red is different dataset in different panels. Specify this in the caption. Please put the correlations in cross-correlation tables as a new right hand column to this figure. At the moment working out which correlation is which, reconciling with the text, and in particular comparing correlation values, is extremely challenging; a table would greatly improve this. (Also, the correlation text at bottom of Fig 3 panels uses different layout to those of Fig 4 panels- very confusing.)
**We have decided not to add other tables of these correlations in order to keep the text succinct. Instead, we have improved the captions for both Fig. 3 and 4, and reduced the text in discussing them to improve clarity, only focusing on the main important points. We hope this will make it easier to follow the conclusions from this table without the need for new tables.**

L242 Context needed: where correlations with obs 1979-present exceed cross-reconstruction correlations 1905-2020, how do cross-correlations compare over the common 1979-2020 period?

**These correlations are higher during the common period 1979-2020, but we have not added this detail to the text in order to try and be more succinct. As mentioned above we have reduced some of the finer details here in the discussion of Figs. 3 and 4 to improve the flow of the paper.**

L242-259 It is hard to follow this discussion of comparison of correlations, partly due to the paragraph structure and partly due to the lack of correlation tables. Indeed I think it is misleading as the text seems to state cross-correlations are all lower than correlations with observations, then later states this isn't true for Abram 2010. To resolve, I'd use a table and address correlations in turn: first Dalaiden with pure proxy-based then Fogt with all other datasets. Also, be sure to sell the take home here; basically none of the cross-correlations are significant pre-1979 so the reconstructions really do tell us very different things!

**Agreed that the previous text was difficult to follow. We have revised this text significantly and removed some individual comparisons that distract from the main points and may seem contradictory.**

L260 'Sudden anomalies': very intriguing in light of current narrative around records. Suggest expanding this, or commenting on implications (e.g. extremes from paleo record should be interpreted with great care- see also end of my previous comment?!)

**For brevity and to improve flow, this sentence has been deleted – it is beyond the scope of this paper to reflect on these individual extremes in any of the reconstructions.**

L273: To aid with seasonal vs annual, compare Fogt SON to Abram et al (2010) ASO?

**Yes, this is done in the right two columns, bottom row of Fig. 4.**

L276 'biased by accumulation at the ice core site': cite?

**We have added the following two references:**
**https://egusphere.copernicus.org/preprints/2023/egusphere-2023-1903/ (Servettaz et al. 2023)**
**10.1029/2018GL081517 (Turner et al. 2019)**

L286: r=0.56 isn't so different from r=0.525, similarly 0.345 vs 0.320! I'd drop line 285-289 to avoid losing the woods for the trees, and slightly rephrase L290 accordingly.

**Agreed, this is a bit too technical and we have removed these lines (including L290).**

Similarly line 305-307 doesn't seem to describe something that's substantially different from the annual? Is line 310-311 (better agreement in seasonal) therefore true?

**We have rephrased these lines to reflect during which seasons the agreement is best across the seasonal cycle, as this is the main point. We have also rephrased lines 310-311 to reflect this main point – not that the seasonal correlations are much better than the annual, but that they are not uniform across the seasons.**

L304 I have a general dislike of calling 'SON' spring for sea ice- it's the maximum! Meterologically correct, cryospherically misleading. Rephrase.
**We agree that this is perhaps not the best term for sea ice, but we were trying to break up using the acronyms JJA, SON etc. continuously in the section. We have changed to 'winter and spring' to 'second half of the year', as a solution.**

Figure 5: Indicate the relevant sector in each panel, to draw the readers eye to where one might expect the correlations to be high and positive.
**Great idea – have added longitudinal boundaries for each dataset in Fig. 5.**

L382 remove bracketed phrase: repetition
**Removed**

Fig 6: The satellite trends look like they start in 1969?
**Well spotted – they did start in 1969 because the previous calculation allowed for up to 10 missing years in each dataset, which was not specified anywhere. To be more conservative, the number of missing years allowed was reduced to 5. This has also been indicated now in the caption of Fig. 6.**

L398 this paragraph is crucial but a bit weak. At L399 (and 414) expansion/recap is needed on why Fig 5 implies role of atmospheric circulation? For me it's more Fig 6 (contrasting trends which we know in obs period are linked to wind patterns) that imply the circulation role? At line 406, Dalaiden pressure is linked to Fogt sea ice, but this seems a bit of a leap of reasoning?
**Yes, we meant Fig. 6 here instead of Fig. 5. We have shortened the paragraph as well to try and improve the main point that the ASL index, known to influence sea ice in the modern era, shows relationships in some of the reconstructions, hinting at the role of the atmospheric circulation here.**

L409: Very helpful paragraph!
**Great, we agree that a summary of key points thus far is helpful and have not modified this paragraph.**

L419: Split this section for readability. 3.2 'Connection to the atmospheric circulation changes' and add at L529 3.3) 'Differences in atmospheric reconstructions'
**Done as suggested, good idea to improve flow.**

L424 'for consistency' with what?
**We were meaning consistency with the sea ice reconstructions, but have decided to delete these words to shorten the text.**

L424-430 over-repetition of metods. Cut down and reference Methods section
**Indeed, we have cut the majority of the text in this paragraph as it is mentioned in the data section earlier.**

L430-433. "Since the Antarctic station pressure reconstructions were generated using a similar statistical technique as the Fogt et al. (2022a) sea ice extent reconstructions, this allows for an evaluation of other estimates that are expected to provide **similar temporal variability** as the Fogt et al. (2022a) sea ice extent reconstructions." Seems at odds with the conclusions that the relationship between Fogt sea ice and Fogt pressure changes over time and with statements at L466- 469. I think this is just me being confused, so please elaborate for clarity of reasoning! **We were aiming to imply that the predictor data were the same in both reconstructions here (not any temporal changes in them with the predictands). However we agree that this was confusing and so have deleted these sentences to improve readability as they are not critical to the main points of the paper.**

L434: "Figure 7 displays the correlations for the Weddell sector sea ice extent from Dalaiden et al. (2021) with the gridded pressure datasets in the top rows, and the Weddell sector sea ice extent reconstructions from Fogt et al. (2022a) with the same gridded datasets in the bottom rows" -> "Figure 7 displays the correlations for the Weddell sector sea ice extent from (top rows) Dalaiden et al, 2021) and (bottom rows) Fogt et al. (2022a) with three gridded pressure datasets"
**Changed as suggested**

L439 'panels' -> 'rows'?
**We have not changed as we meant each individual figure panel, not across a row.**

L456-L461: (overly lengthy) Change to : "However, **for** the Fogt and Connolly (2021) pressure dataset, the relationship **with** Weddell sea ice extent, regardless of the sea ice extent estimate (Dalaiden et al. (2021) or Fogt et al.(2022a)) shows a change in **sign** poleward of 60°S in the 1905- 1978 period".
**Changed as suggested.**

L471: 'this pattern' what pattern?
**We have rephrased this paragraph substantially and merged with the following one to clarify our main point and improve flow.**

L503: clarify Orcadas is met data not sea ice.
**Sentence has been rephrased to clarify pressure observations at Orcadas.**

L511- 'change sign' but neither sea ice time series has significant correlation with Orcadas SLP so I think not much can be read into this.
**We have softened this language by starting this sentence with 'Although usually not statistically significant',...**

L512: 'spatial pressure change' -> 'change in the Weddell SIE-gridded SLP correlation change'
**Changed as suggested**

L516 and other locations: add explanation of why analysis is split at 1945.
**For this case, 1945 just referred to the significance in Fig. 7. We have changed from 'starting prior to 1945' to 'in the early 20$^{th}$ century' to clarify it wasn't an arbitrary date. The only other notable split at 1945 is for Fig. 12, which was chosen as the first 5-year period when observations first became available at Faraday (Faraday observations start in 1947).**

L517 'weakly correlated' –> 'have no statistically significant correlations'
**Changed as suggested**

L538-552: (presentation) Overly lengthy, repeats content from methods, and implies slightly that new sensitivity runs were performed for current paper. Suggested succinct reword: 'Further, we analyse the single-proxy-based reconstructions of O'Connor (2023) to pinpoint the sensitivity of these correlations to using only tropical (coral) or Antarctic (ice core) paleo constraints'. Also this is mostly a repetition of methods section.
**Agreed that it is repetitive of methods section, so have decided to remove these sentences to improve flow.**

L552 New paragraph. Re-explain why 1956 break point?
**Have stated 'Prior to the start of most Antarctic pressure observations,...' at the start of the new paragraph. This break point is more relevant for Fig 13 than here (when Antarctic observations are plotted), but kept the same time period for consistency.**

L552-L602: This section is hard to follow and needs more careful wording/separation of 1) discussion of what's driving the patterns in the reconstructions (comparing panel patterns), 2) how well they are correlated with Fogt and Connolly and 3) what we can learn (i.e. what's driving variability in Fogt and Connolly? Or is Fogt and Connolly considered as more reliable?) Is one possible interpretation that Fogt and Connolly is biased in East Antarctica due to over-dominance of tropical here in this reconstruction? Specifically the word 'agreement' is ambiguous e.g. at line 569 'agreement improves' –agreement between assimilation reconstructions, or with Fogt and Connolly?
**We have revised the text in these lines substantially, and broke it into two separate paragraphs. We have also removed unimportant details that can distract from the main points. It is not possible to determine which is more reliable based on these comparisons, and if Fogt and Connolly is biased due to tropical teleconnections, this would more like emerge in the Pacific sector, not around East Antarctica.**

L556: I think this implies O'Connor 2021 has forced prior but that L456 implies it has fixed?
**See our previous response to your comment. Have also deleted ' fixed prior' on L456.**

L563-564: rephrase to 'not surprising that the coral-only dataset of O'Connor et al is better correlated with F&C'- but this is only true pre-1957?
**We have removed this sentence to improve flow.**

L566- 'is opposite that of... especially in East Antarctica' I'd rephrase as 'except in the Amundsen- Bellingshausen sector' and then include the note about teleconnections being less important for East Antarctica.
**We have shortened this sentence to improve the flow, slightly different than suggested, but still reference the reduced teleconnections in East Antarctica.**

L581-L582: belongs in methods
**Deleted here and added to methods.**

L592 'agreement' -> 'correlation'
**changed as suggested**

L619: 'changes' (implicitly in time) or 'differences' (between datasets)? Also L619 "change in correlation in Fig 12" what's meant here?
**Rephrased this to now read: 'The changes in linear trends through time across the various datasets ( ) are also noteworthy in Fig. 12'**

Fig 12: Again consider correlation tables. Which value refers to which dataset is very hard to interpret. The Fogt dark green and obs black are almost indistinguishable. What's the LHS?
**We have added detail to the legend that the statistical values for both correlation and trends are color coded following the timeseries colors in the legend, and added this detail to the main text as well, in order to avoid adding tables that would increase the length of the paper. The FC21 and observations are indeed indistinguishable in almost in any color pattern as there is very strong agreement between the two (higher correlations and trends) – the FC21 was calibrated to match these data.**

Text (L608) 'correlations with obs improve after 1945' implies it is correlation with obs but text in panels and the fact most obs aren't available until 1945 implies not.
**Have reworded to indicate that the correlations with observations are generally higher than the correlations with FC21 (and removed the time reference to 1945).**

L641 'statistically significant differences in the underlying atmospheric circulation' are there?
**We have deleted this sentence.**

L660: Give Vostok longitude.
**Have now added latitude and longitude for Vostok in the text.**

L678: The word 'incorrectly' is important; it implies Fogt and Connolly early C20 trends are in fact incorrect, which sheds information on which of all the reconsturctions is more reliable. Discuss.

**This is very complex, as all datasets are 'incorrect' somewhere, and specifically in terms of agreement with the positive trends at Vostok in East Antarctica, the coral-only is the only match here, but this dataset misses the significant negative trends along the Antarctic coastline. We have rephrased this and removed the word 'incorrect'. Each dataset has its own strength and weaknesses and none capture the pattern in observations fully, which we hope is clearer in the revision.**

L693: 'reverse' when? Previous sentence is only discussing pre-1956 but trend reversals are later?

**Clarified the reversal in trends in the Fogt and Connolly (2021) dataset throughout the 20th century.**

Discussion: Byrd data limitations should be mentioned briefly in results section.

**Added a sentence about this in the methods, but have also kept some in the discussion shortly as it is an important point for future work.**

L778: "not able to determine the validity of changes in the atmospheric circulation in the early to mid 20th century." ... and therefore not able to deduce which sea ice reconstructions are the best representation of reality?

**Yes, more work to be done as indicated in the next several sentences!**

L782-> (on strengths and weaknesses and lessons for users): I wonder if this could be recap some key specific conclusions from the paper? Or if the authors feel uncomfortable doing this, flag the key questions that remain to be answered to understand which datasets are more reliable.

**The specific conclusions were recapped in the previous paper, and this paragraph is essentially talking about the inability to fully understand early 20th century atmospheric circulation around Antarctica – the key unknown question this paper leaves unanswered (but implores for future work) – to better know which datasets are more reliable.**

Technical Comments
L18 minima -> lows
**Changed**

L20 add 'since 1900'
**done**

L27 'several'-> '5'
**changed to 'Five'**

L113 remove 'various'. Put this sentence to end of para.
**changed as suggested**

L125 link failed but https://rda.ucar.edu/datasets/ds570-0/ worked
**fixed, thanks**

L151 'combines of' -> 'combines'
**Done**

Table 1 column 2 heading 'type' sufficient. In column 4 also ref Figs 2b) and c) where relevant
**shortened column 2, refs in column 4 to Figs 2b and 2c were already present**

Table 1 last row/column 'interpolated' typo
**Fixed**

L174 Remove 'a' before 'three'
**fixed**

L219 with->of
**Sentence has been rephrased, revision no longer relevant**

L224 ', and the Thomas and Abram...' -> '. The Thomas and Abram...'
**Sentence has been deleted**

L242: Remove 'however'
**deleted**

L246 '; part'-> '. Part'
**changed**

L264 (end): 'created in'->'created with'
**Changed**

L279 'Compared to' -> 'in contrast with'
**changed**

L428 'A blend interpolated' -> 'a blend of interpolated'
**sentence has been deleted**

L553: 'between' -> 'for' (authors are not cross-correlating these datasets with each other)
**changed**

Fg 12 Be explicit that point A = panel a)
**made explicit in the figure caption**

L689 'with' -> 'between'
**Changed**

L691 repeated '1905-1956'
**Sentence has been revised in response to other reviewer comments**

L791 'Ceter'->'Center'
**fixed**

---

## Author Comment (AC2)

**We thank both reviewers for their comments and detailed feedback that have helped to improve the manuscript significantly. In response to both reviews, we have shortened the text as much as possible, and have used acronyms to refer to the various datasets we analyze in the paper. Although we have not added the tables suggested by reviewer 1, we have made it clearer in the text and figure captions that the correlations in each figure are color coded, which will hopefully clarify their values; we thought that adding other tables would not only increase the length of the paper, but also potentially add to the challenge of overly detailed and distracting text that was present in the first draft.**

**Our responses to the reviewer comments are in bold below.**

**Reviewer 2**
**A Comparison of South Pacific Antarctic Sea Ice and Atmospheric Reconstructions Since 1900**
The study compares different proxy-based and station-based Antarctic sea ice extent reconstructions throughout the 20th century with the aim to investigate the reason behind their discrepancies, with a particular focus on atmospheric circulation. The reconstructions show a good agreement in the Weddell sea, and a weaker agreement in the Ross and Bellinghausen seas possibly due to changes in atmospheric circulation. The sea ice reconstructions are compared with pressure reconstructions spanning the Southern Hemisphere, finding a connection of mid-latitude atmospheric circulation with Antarctic sea ice variability.
The study is very well detailed and definitely worth publication. I particularly appreciated that years of efforts in reconstructing sea ice using paleoclimate archives are being used in a wide spatial and temporal context to investigate underlying climate relationships. There are a few typos here and there but otherwise it is excellently written, with a good flow and level of English. However, while I appreciate the level of detail of the paper, I think it could have been more concise, in order to facilitate the reader to go through the very long text. I suggest to try to simplify the text whenever possible as I had myself to read the paper several times to understand the reasoning of the authors. One possibility could also be to report a discussion of the results in the "Discussion" section that follows the same structure as the "Results" section, to better synthesize the main findings of the article.
Also, as the authors state on page 15, the various sea ice reconstructions represent different aspects of sea ice variability and moreover are based on different methodologies and on a different number of records considered. I suggest to be more clear of these limitations in the text when examining the different sea ice reconstructions and their relationship with atmospheric circulation.
**We thank reviewer #2 for their comments and overall positive reflection on this paper. We have made substantial revisions to the text in response to both reviewers that hopefully have improved the flow and shortened the text, which should ultimately make it easier to understand the main points. We have also added text to discuss more of the limitations of each dataset throughout.**

Please find below some technical comments:

Table 1: in the Abram et al. (2010) and Thomas and Abram (2016) rows I would indicate the type of sea ice proxy used. In the Dalaiden et al. (2021) row please modify "isotope-enable CESM1" with "isotope-enabled CESM1"
**Done**

Figure 2a: please change the title to Fogt et al. (2022a) for consistency
**Done here and throughout Figs. 2-8.**

Line 151: "The CDS algorithm output combines ice concentration". Remove "of".
**done**

Line 169: "sediment-derived"
**Done**

Line 172: "ice core-derived"
**Done**

Line 174: remove "a" from "used a three ice cores"
**done**

Line 176: "Further West" with capital letter
**Have not changed as West is not a proper noun in this case so no need to capitalize**

Line 190: "and those variables given by the Earth System Model…" please specify which variables
**Added**

Line 194: the Dalaiden et al. (2021) reference is repeated twice, please remove one
**Not changed – couldn't find repeated reference**

Line 216: please change with "paleo data, including those from data assimilation -based reconstructions"
**Changed as suggested**

Figure 4: please make clearer that the middle and right columns refer to Abram et al. (2010). Why not change the colors from Thomas and Abram (2016)?
**We have kept the red throughout as the paleo-based ice core reconstruction (as in Fig. 3) but have indicated the right columns refer to Abram et al. (2010)**

Line 306: "Dalaiden et al (2021) are typically highest in JJA". Please remove "in"
**Done**

Figure 5: I suggest to indicate the sectors in the figure as in figure 1a green line. Again, it is not clear which Fogt et al. (2022) paper is referred to.
**done**

Line 344-357: Here the authors compare the Abram et al. (2010) sea ice reconstruction with the other reconstructions in figure 5. However, such reconstructions are based only on three ice core records. I suggest to mention here that this is one of the limitations of that reconstructions with respect to e.g. Dalaiden et al. (2021) which is based on several records.
**Have added a sentence reminder reader of this fact at the end of this paragraph**

Line 368: typo in "geopotential height"
**fixed**

Line 391: "opposite to the strong positive trends". Please add "to".
**added**

Figure 7: as in Figure 2a please change the title to Fogt et al. (2022a) for consistency
Figure 8: Again clarify which Fogt et al. paper
**Done for both figures as suggested.**

Line 510: "it is not surprising that the relationship". Remove "then" and "both"
**This sentence has been revised in response to reviewer 1**

Figure 10: Please rename figure titles
**Not clear what the reviewer wants the titles renamed to, as it is explained what each row is in the figure legend.  Have not changed.**

Line 566: please add "to" after "opposite"
**Done**

Line 573: "This suggests that the Fogt...". Please add "that"
**done**

Line 591, 593 and 594: please check the figure number, it should be figure 11
**well spotted, changed all**

Line 608: Please refer to the figure
**added**

Line 616: I suggest to change "hence why the" with "explaining why the"
Line 617: add parenthesis before "black correlation"
**this text on both lines has been deleted**

Line 741: Figure 12 instead of 11
**Thanks, changed**